# In vivo localization of chronically implanted electrodes and optic fibers in mice

Bálint Király [1,2,3], Diána Balázsfi[1], Ildikó Horváth [2], Nicola Solari [1], Katalin Sviatkó[1,4], Katalin Lengyel[1], Eszter Birtalan[1], Magor Babos[5], Gergő Bagaméry[5], Domokos Máthé[2,6], Krisztián Szigeti[2] & Balázs Hangya [1✉]

Electrophysiology provides a direct readout of neuronal activity at a temporal precision only limited by the sampling rate. However, interrogating deep brain structures, implanting multiple targets or aiming at unusual angles still poses significant challenges for operators, and errors are only discovered by post-hoc histological reconstruction. Here, we propose a method combining the high-resolution information about bone landmarks provided by micro-CT scanning with the soft tissue contrast of the MRI, which allowed us to precisely localize electrodes and optic fibers in mice in vivo. This enables arbitrating the success of implantation directly after surgery with a precision comparable to gold standard histology. Adjustment of the recording depth with micro-drives or early termination of unsuccessful experiments saves many working hours, and fast 3-dimensional feedback helps surgeons avoid systematic errors. Increased aiming precision enables more precise targeting of small or deep brain nuclei and multiple targeting of specific cortical or hippocampal layers.

[1] Lendület Laboratory of Systems Neuroscience, Institute of Experimental Medicine, Budapest, Hungary. [2] Department of Biophysics and Radiation Biology, Semmelweis University, Budapest, Hungary. [3] Department of Biological Physics, Eötvös Loránd University, Budapest, Hungary. [4] János Szentágothai Doctoral School of Neurosciences, Semmelweis University, Budapest, Hungary. [5] Mediso Medical Imaging Systems Ltd., Budapest, Hungary. [6] CROmed Translational Research Centers, Budapest, Hungary. ✉email: hangya.balazs@koki.mta.hu

One of the most common approaches of modern experimental neuroscience is recording or influencing the activity of small brain areas or specific cortical layers in live laboratory animals. For instance, electrophysiology provides a direct readout of neuronal firing at the single-cell level with high temporal resolution[1–3]. In addition, the recent advent of optogenetics allows activating or suppressing genetically defined groups of neurons via the cell-type-specific expression of light-gated ion channels and light delivery via optic fibers[4,5]. The combination of the two enables optogenetic tagging, i.e., light-guided assessment of neuron identity during extracellular recordings from awake-behaving animals[6–9]. Finally, fiber photometry techniques are on the rise, requiring fiber-optic implants similar to optogenetic experiments[10–13].

Currently, there is strong focus on engineering ever-evolving opsin actuators, optimizing viral vectors for their delivery and improving recording electrodes in terms of channel count and arrangement, tissue damage, and durability[5,14–21]. However, experimental success rates are still largely limited by operation and targeting techniques, which remained essentially unchanged since the introduction of stereotaxic surgeries.

Mice represent an attractive model due to their relatively low price, short generation time, genetic tractability, complex behaviors, and a level of reported homology between rodent and primate brains. However, surgeries on the mouse brain have gradually become more demanding and error-prone as the field moves toward more complex experiments, aiming at small and often multiple targets simultaneously, hitting the same target multiple times (e.g., virus injection and optic-fiber implantation), or implanting at unusual angles to avoid large blood vessel. Targeting deep-brain nuclei or specific cortical or hippocampal layers poses a significant challenge, as slight deviations from ideal implanting directions or coordinates often lead to missing circumscribed targets even in the hands of expert operators. Errors are only discovered at the end of the experiments by postmortem histological reconstruction, often resulting in months of wasted experimental work and lab resources, especially when behavioral training is involved.

In order to improve the efficiency of these experiments, we developed a new procedure in mice inspired by techniques used in human deep-brain surgery and brain radiation therapy[22–25]. CT imaging has been used to localize 50-μm-diameter electrodes postmortem with an accuracy sufficient for larger (>1000 μm) structures of the rat brain[26] and large (200-μm-diameter) electrodes and lesion sites were localized with CT and MRI in rats in vivo[27]. However, the twofold size difference between mouse and rat brains and the need for targeting small nuclei makes sufficiently detailed imaging and localization challenging, and do not allow direct implementation of these methods for precisely localizing small-diameter implants in the mouse brain. To overcome these limitations, we used high-resolution micro-CT imaging that allowed accurate measurements of tetrode electrodes, silicon probes, and optical fibers with respect to bone landmarks. Improving the resolution and the signal-to-noise ratio (SNR) of CT images may be achieved at the expense of increased radiation dose. Therefore, we optimized imaging settings and determined a safe parameter regime that allows precise in vivo localization without adverse health effects.

However, CT does not provide soft-tissue contrast, making it difficult to judge whether the implant is in the target area. Therefore, we performed structural MRI scans that provide excellent soft-tissue contrast and merged them with the micro-CT images, similar to human surgery techniques. The procedure consists of preoperative and postoperative imaging, which is followed by the registration (i.e., aligning) of the images with an atlas coordinate system (Fig. 1), providing localization of implants

with a precision that matches post hoc histological reconstruction. This noninvasive in vivo localization technique enables verifying the success of implantation directly after surgery, and increases the efficiency of the experiment by (i) adjusting the recording depth to reach the target, (ii) terminating in the case of mistargeting or other problems, and (iii) systematically improving surgery skills by providing fast feedback. We estimate that this method may save up to an order of 1000 working hours per chronic recording/optogenetics/fiber photometry projects, depending on the difficulty of the surgery and the length of behavioral training or other repeated procedures.

## Results

**CT-based implant localization.** High-resolution CT images of the skull allow image registration to a common stereotaxic coordinate system based on bone landmarks. As important bone landmarks are often masked by surgically applied radiopaque dental adhesives, these scans are ideally performed in intact animals. Therefore, we conducted preoperative micro-CT measurements of anesthetized mice fixed by the top incisor teeth in a specialized isoflurane mask (similar to ref. [28]) to avoid small movements and displacements (Fig. 2a). Preoperative micro-CT imaging was performed at 35-μm resolution that allowed sufficient visualization of skull sutures and other bone landmarks at low radiation dose (see "Optimization of image quality and radiation dose" below). Next, we aligned the imaging planes with canonical 3D axes used in brain atlas coordinate systems, similar to stereotactic adjustment of skull position, often referred to as "leveling" (Fig. 2b). Leveling of the sagittal plane was based on Bregma and Lambda points on the surface of the calvaria, which were defined as the midpoints of the best-fit curves on the coronal and the lambdoid sutures, respectively[29]. We used a collection of symmetrical bone structures to aid leveling of the horizontal and coronal planes, including the squamous, petrous, and tympanic part of the temporal bone, the zygomatic process, the temporal line, the basisphenoid bone, the presphenoid bone, the tympanic bulla, the ear canal, inner ear structures, and other symmetrical points of the cranium (yellow dashed lines in Fig. 2b and Supplementary Fig. 1).

Standard stereotaxic surgeries were performed by experienced operators to implant chronic electrode drives housing eight tetrodes and an optic fiber. During surgery, the horizontal limb of the diagonal band of Broca (HDB, $n = 5$), a deep-brain target in the basal forebrain area (+0.74 mm anteroposterior, 0.6 mm lateral, and 5.0 mm dorsoventral from Bregma, see "Methods"), or the medial septum of the basal forebrain (MS, $n = 2$) was targeted (+0.9 mm anteroposterior, 0.1 mm lateral, and 3.9 mm dorsoventral from Bregma). MS is a midline structure; hence, it was implanted at a 14° angle with respect to the vertical axis in the coronal plane to avoid the sinus sagittalis superior.

To accurately localize the implanted electrodes, we performed postoperative CT scanning at 19-μm resolution after a 4–12-day recovery period using the same procedure as detailed above (Fig. 2c). To minimize shadow artifacts introduced by metal objects in the region of interest (ROI) of the CT images, we positioned the metal parts of the drive, i.e., wiring and pins of the electrode interface board and occasional hypodermic tubing or head bars, to point away from the approximately coronal plane in which the X-ray source was rotating. Next, the postoperative CT scan was co-registered with the preoperative one that had already been aligned to the stereotaxic coordinate system by matching the corresponding bone structures using rotation and translation operations on the postoperative image (Fig. 2c). Finally, the implant was segmented with an intensity threshold applied on a whole-brain ROI (dark-blue area in Fig. 2c). As a result, we were

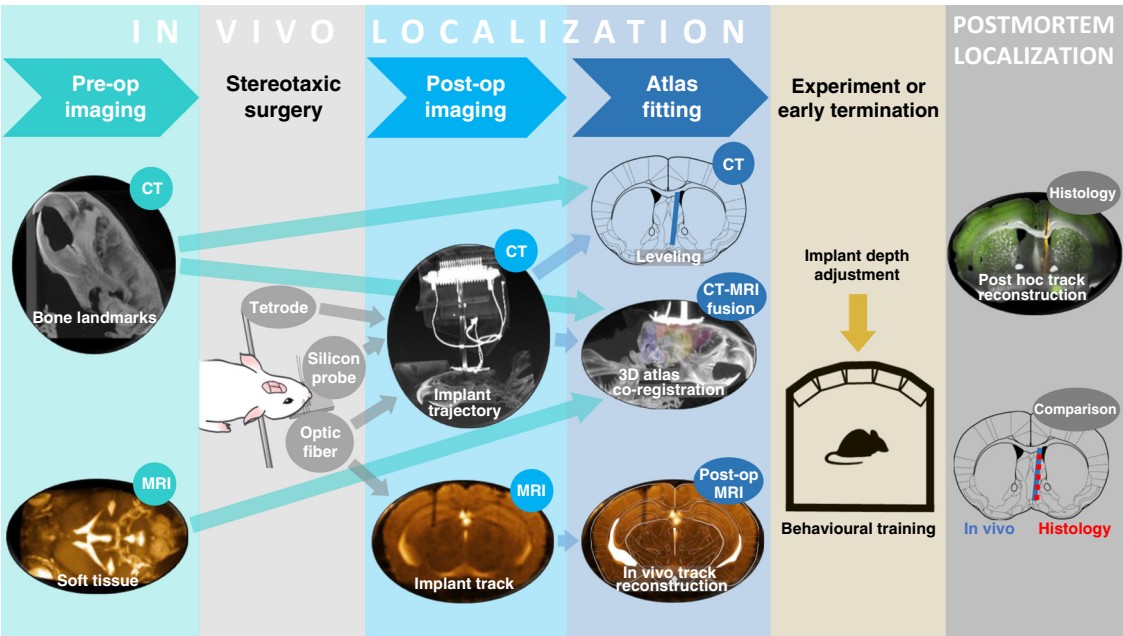

**Fig. 1 In vivo localization workflow.** A few days before the surgery, preoperative CT and MRI scanning is performed, providing anatomical information about bone landmarks and brain structures, respectively. After 4–12 days of recovery from the stereotaxic surgery, postoperative CT images are taken, and the position of the implant is determined in a co-registered atlas coordinate system. In case of nonmetal implants, a single postoperative MRI scan provides an alternative option, which allows in vivo track reconstruction analogous to histological track reconstruction. Based on the results, the trajectory of the implant is extrapolated, the depth of the implant may be adjusted before the experiment, or in the case of mistargeting, early termination is considered. After the experiments, the accuracy of the in vivo localization is verified by post hoc histology.

able to read the stereotaxic coordinates of the tip of the implant relative to the Bregma referenced to brain atlas coordinates (Paxinos and Franklin's, the Mouse Brain in Stereotaxic Coordinates[29]) and calculate the angle of the implant trajectory. We used trigonometric functions to calculate the future position of the implant along its projected trajectory.

**Localization of multiple-targeting implants, silicon probes, and optic fibers.** Questions about information transfer and coordinated processing across distant brain areas are increasingly coming into research focus; hence, targeting multiple brain areas simultaneously is becoming common practice. Therefore, we tested whether our localization technique could be applied to implants with multiple targets. To this end, we implanted a custom-built dual-electrode drive housing eight–eight tetrodes with one–one optic fiber at a fixed distance of 3.85 mm for simultaneous targeting of two subcortical neuromodulatory centers, the HDB of the basal forebrain and the midbrain ventral tegmental area (VTA, −3.1 mm anteroposterior, 0.6 mm lateral, and 4.05 mm dorsoventral from Bregma). The presence of more metal parts in these drives introduced somewhat more noise in the postoperative CT images compared to single tetrode-bundle implants. Nevertheless, we were able to reconstruct both tetrode trajectories and localize the tips using the in vivo technique (Fig. 3a).

Although tetrode recordings may be considered the mainstay of in vivo extracellular electrophysiology, a large variety of silicon probe electrode designs are gaining popularity due to their high channel count, customized contact site configuration, and ease of use. Therefore, we tested the in vivo localization technique on different types of silicon probe implants (n = 3, Buzsaki probe, 52-μm width and 15-μm thickness; polytrode probe, 113-μm width and 15-μm thickness; edge probe, 150-μm width and 15-μm thickness) lowered into the dorsal or ventral hippocampus (see "Methods"). The probe trajectories were successfully

visualized by micro-CT imaging, demonstrating that the localization technique is readily applicable to silicon probe electrodes (Fig. 3b). Silicon probes get thinner near their tips, and it is challenging to assess tip position with histology. To verify that we detected the tips accurately with micro-CT imaging, we measured the total length of the probes from base to tip in the CT images and compared this image-based estimated length with nominal dimensions published by the provider (Supplementary Fig. 2). We found negligible mean difference between the two measures (mean ± standard deviation, 23 ± 5 μm), confirming that the most ventral point of the radiodense area is a precise estimate of tip location.

We have demonstrated precise localization of metal electrodes using their high-contrast signal in the CT images. With the rapid progress of optical methods for imaging and manipulating neural activity, including optogenetics and fiber photometry, optic-fiber implants have become common in chronic in vivo experiments. Based on the approximate Hounsfield unit (HU) of different glass and plastic objects[30,31], silica glass and plastic optic fibers are expected to provide a signal intensity between the ones induced by bone structures and soft tissue. We implanted mice (n = 3) with a 105-μm core (250-μm outer diameter) multimode silica optic fiber targeting the HDB, the VTA, or the dorsal hippocampus, suitable for stimulating or inhibiting neural activity in optogenetic experiments, and subsequently tested the localization method. Although the nonmetallic optic fibers indeed provided a lower-contrast signal compared to the metal electrodes, micro-CT imaging proved to be sufficient to reveal the position of single optic fibers on the soft-tissue background (Fig. 3c).

**Implant localization based on CT–MRI fusion.** Previously, we registered the tips of the implants to stereotaxic atlas coordinates based on the combination of pre- and postoperative CT scans. However, as CT provides practically no soft-tissue contrast in the

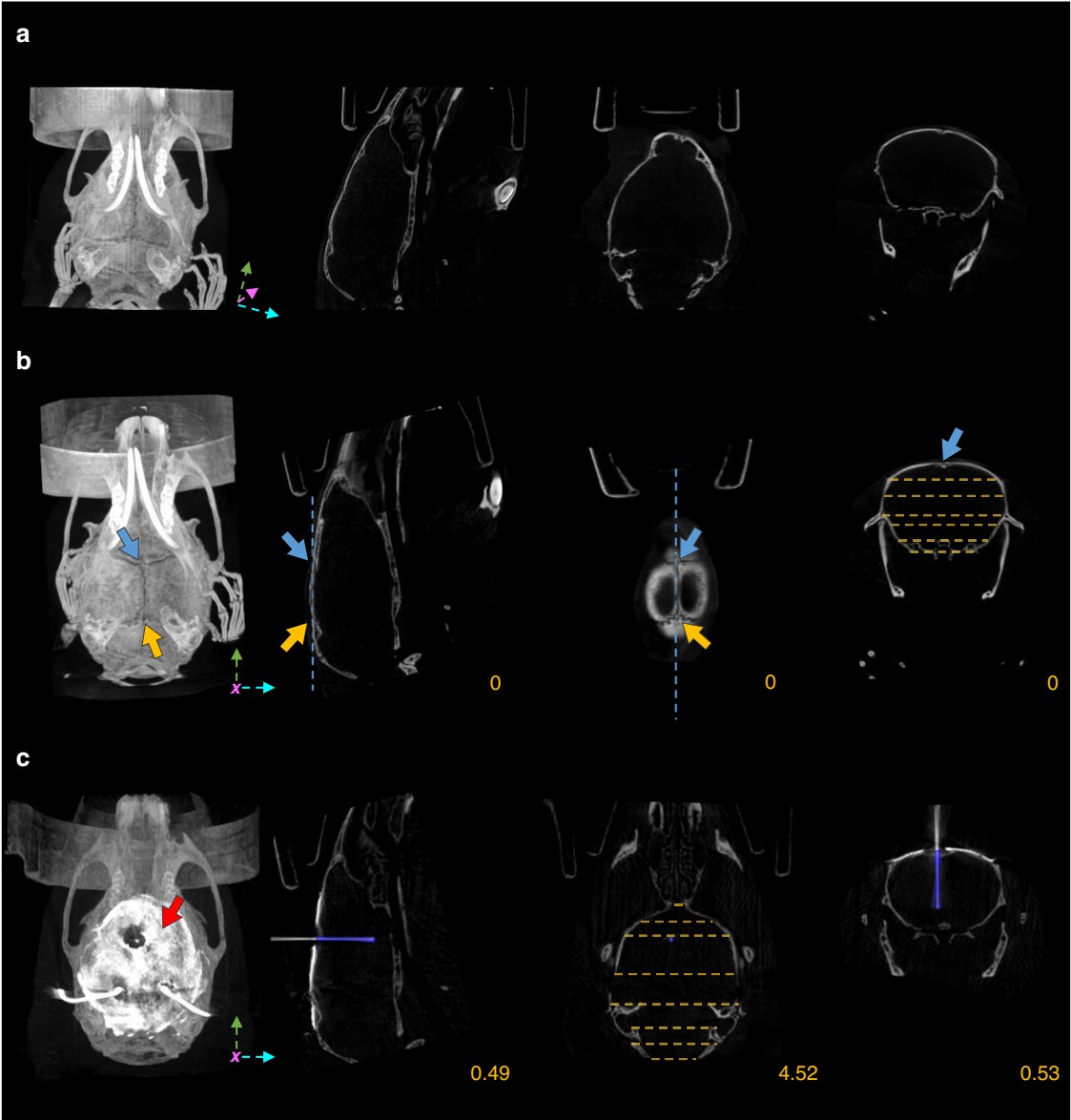

**Fig. 2 CT-based implant localization. a** Raw preoperative CT image. Canonical 3D axes used in brain atlas coordinate systems (green, pink, and cyan arrow) are not aligned with the coordinate system of the CT image. **b** The preoperative CT image was aligned to the atlas coordinate system by leveling the Bregma (blue arrows) with the Lambda (yellow arrows) in the sagittal plane (blue dashed lines) and symmetrical bone structures in the coronal and horizontal planes (yellow dashed lines). **c** After the implantation of an eight-tetrode microdrive, postoperative CT imaging was performed, and the scan was co-registered with the preoperative CT image. The skull was covered with a radiopaque dental adhesive (red arrow), which hid Bregma and Lambda. The implant was segmented (blue area) with an intensity threshold applied on a whole-brain region of interest (ROI). CT images are shown in grayscale. Images in each row from left to right, maximum intensity projection, sagittal slice, horizontal slice, and coronal slice. Yellow numbers refer to the corresponding mediolateral, dorsoventral, and anteroposterior stereotaxic coordinates in mm from Bregma.

mouse brain, it is hard to judge the area localization of the implant as well as the distance from neighboring structures in three dimensions based on the images. In addition, CT-based registration relies on a few bone landmarks without any possibility of correction based on brain structures. Consequently, the above method cannot take individual variability of brain morphology into account, and strongly depends on the somewhat subjective identification of the Bregma point.

Therefore, we further improved our localization method, based on the CT–MRI fusion technique routinely applied during human deep-brain stimulation (DBS) surgeries and brain radiation therapy[22–25]. In addition to the preoperative CT, we performed a fast $T_1$-weighted preoperative MRI scan directly after the CT scans. To aid the subsequent CT–MRI fusion, mice were kept in the same scanning bed across the two modalities, avoiding postural changes and large disparities between the imaging planes. Next, we developed a multistep procedure to co-register our scans with a 3D brain atlas[32], enabled by the soft-tissue information obtained from the preoperative MRI (Fig. 4). As an important rule to guide precise image registration, in each of the following steps, either images from the same modality or images from the same animal acquired in the exact same position were co-registered.

(i)  The preoperative CT image was aligned to stereotaxic axes as described for CT-based localization.

(ii)  The preoperative CT and MRI images were co-registered (Fig. 4a). This could be achieved in a relatively simple

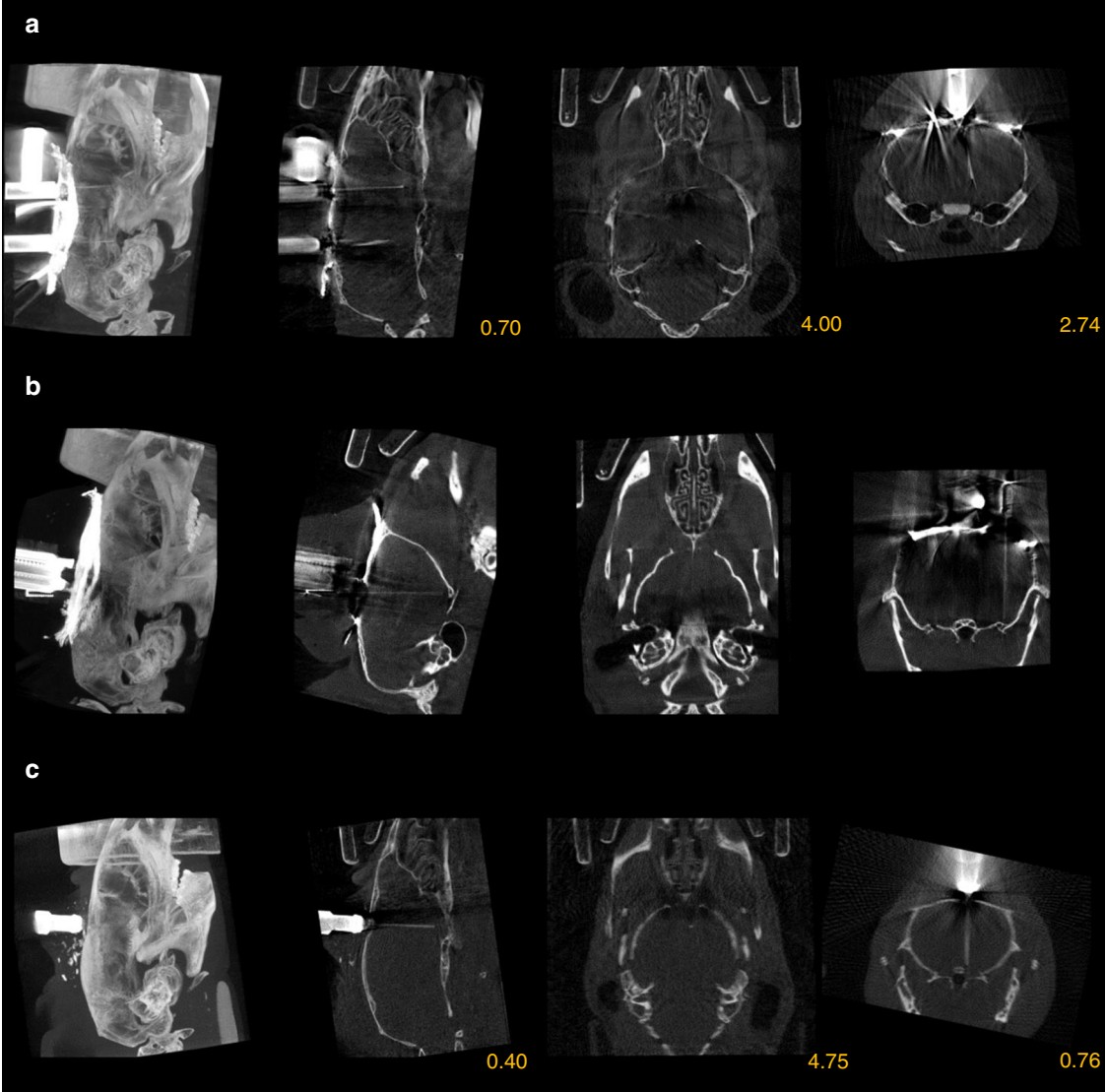

**Fig. 3 Localization of multiple-targeting implants, silicon probes, and optic fibers. a** CT image of a dual-tetrode drive implanted in the HDB and the ventral tegmental area (VTA). **b** CT image of a single-shank Buzsaky-type silicon probe implanted in the ventral CA3. The image was rotated to visualize the thin silicon probe. **c** CT image of an optic fiber implanted in the HDB. CT images are shown in grayscale. Images in each row from left to right, maximum intensity projection, sagittal slice, horizontal slice, and coronal slice. Yellow numbers refer to the corresponding stereotaxic coordinates in mm relative to Bregma (except for panel **b**, where the image was rotated for visualization purposes).

fusion procedure involving small translations applied on the MRI image, since the two scans were acquired in the same scanning bed while keeping the position of the head of the animal fixed. The transformations were based on the skull, which provided high-contrast signal in CT images while appearing as a distinct lack of signal between soft-tissue structures in the MRI.

(iii) The preoperative MRI image was co-registered with the qualitatively best-matching sample MRI image of the 3D atlas, based on the contour of the brain and the position of the ventricles in multiple planes. This was achieved by a combination of translations and affine transformations, the latter of which ranged up to a few percent. Note that this was the only step in the protocol that required a non-Euclidean transformation. If entirely uniform fitting was not achievable across the whole brain, we paid special attention to the perfect alignment of the structures close to the area of interest, i.e., the surgical target area.

(iv) We applied the transformation matrix that realized the best alignment in step (iii) on the MRI atlas, thus obtaining registration of individual mouse MRI scans to a reference MRI atlas (Fig. 4b). Thus, the first four steps resulted in the co-registration of the preoperative CT and the MRI atlas, providing individualized brain area information in a stereotaxic coordinate frame aligned to bone landmarks.

(v) We verified the fitting of characteristic bone landmarks such as the Lambda and the sinus frontalis with the contour of the atlas, and in the case of misalignment, the first four steps were revised.

(vi) The postoperative CT image was co-registered with the preoperative CT as described for CT-based localization, resulting in a fusion between the postoperative CT and the reference MRI atlas (Fig. 4c). In the coronal slices containing the tip of the implant, the corresponding coronal plane of the anatomically more detailed and readily scalable Paxinos atlas was fitted on the three-dimensional

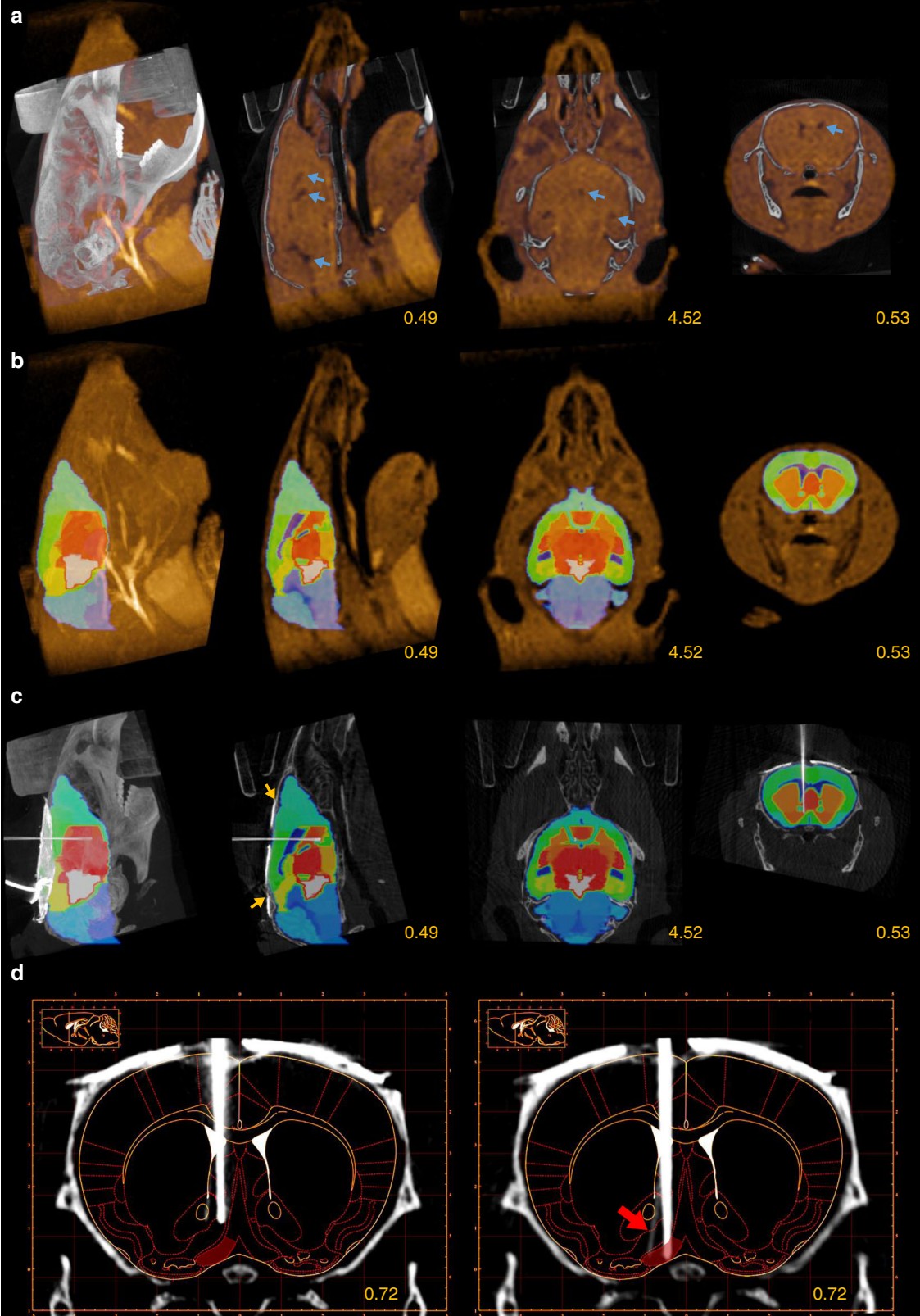

**Fig. 4 Implant localization based on CT–MRI fusion. a** Preoperative CT and MRI images were fused (same mouse as in Fig. 2). Blue arrows show the ventricles on the MRI image used for atlas co-registration. Images from left to right, maximum intensity projection, sagittal slice, horizontal slice, and coronal slice. Yellow numbers refer to the corresponding stereotaxic coordinates in mm relative to Bregma. **b** A 3D MRI atlas was co-registered with the MRI image. **c** The 3D atlas was co-registered with the postoperative CT image. Yellow arrows show Lambda and sinus frontalis. **d** The corresponding coronal plane of the Paxinos atlas was fitted on a coronal slice containing the tip of the implant after surgery (left) and at the end of the electrophysiology experiment (right). The figures clearly show that the implanted tetrodes went through the target area (HDB, red shading) during the experiment. The red arrow shows a tetrode that separated from the bundle. CT and MRI images are shown in grayscale and goldscale, respectively.

atlas (Fig. 4d), which aided further quantitative comparisons (see below).

This procedure resulted in a three-dimensional image displaying the trajectory of the implant in the brain. Similar to the planning phase of human DBS surgeries, we could determine the exact position of the implant relative to the target area and predict the future positions along the planned descent. This allowed more precise knowledge of the recording position throughout the entire experiment, and—in the case of mistargeting—provided a chance to determine the exact source of error (see below).

**Localization of hippocampal implants using high SNR MRI imaging.** In the previous section, we demonstrated that preoperative MRI images can be used for atlas co-registration based on the position of the ventricles, and thus aid the in vivo localization of deep-brain implants. Superficial structures are easier to hit than deep-brain targets; however, the precise position of the implant in the area, especially the depth in layered structures, is crucial to most experiments. For neuroscience applications that require implanting neocortical or hippocampal structures, MRI images acquired with higher field strengths ($\geq$3 Tesla) can provide a detailed visualization of the structure. This may further help atlas fitting by matching the position of local soft-tissue structures, such as the corpus callosum, the hippocampal fissure, or the cortical surface, allowing an improved layer localization of neocortical and hippocampal implants.

To test this, we implanted the dorsal hippocampus ($n = 3$, two mice with silicon probes, one with a 105-µm core optic fiber) and took high SNR $T_1$-weighted preoperative MRI images in addition to the previously described pre- and postoperative CT images (Fig. 5a). By visualizing the hippocampal formation in addition to the ventricles, these images allowed registration to the MRI atlas (step (iii) above) paying special attention to the area around the implant (Fig. 5b and Supplementary Fig. 3). As a result, we were able to precisely localize the implant's trajectory in the hippocampus; for silicon probe implants, this also allowed us to reconstruct the layer-specific position of each contact site based on its nominal distances from the implant tip (Fig. 5c). Localizing the trajectory might also allow the precise adjustment of the probe with suitable microdrives[33].

In addition, we tested whether nonmetal optic fibers can be directly localized with high-quality postoperative MRI scans. We implanted the dorsal hippocampus in two mice bilaterally, with a 105-µm core and a 50-µm core optic fiber on the two sides. We took $T_2/T_1$-weighted high-contrast MRI images with the isovoxel resolution set to 120 µm, directly visualizing the optic-fiber tracks in the tissue, along with anatomical landmarks, including the corpus callosum, the hippocampal fissure, and the ventricles (Fig. 6a). This allowed us to fit the histological atlas directly on the MRI images and reconstruct implant tracks analogously to post hoc histological localization, with a precision only limited by image resolution (Fig. 6b and Supplementary Fig. 4). Finally, a more detailed $T_2$-weighted MRI image of the hippocampal formation could be acquired by collecting data from a thick (600-µm) coronal slice (with 100-µm in-plane resolution), allowing us to precisely measure the depth of the implant tip relative to the hippocampal fissure (Fig. 6c).

**Quantification of localization accuracy using gold-standard histology.** Postmortem histology and anatomical reconstruction of the implant's track is considered as gold standard technique for implant localization. We tested the accuracy of in vivo localization against this gold standard by examining the outcome of $n = 12$ implantation surgeries ($n = 9$ tetrode implants in eight mice targeting the HDB ($n = 6$), the MS ($n = 2$), and the VTA ($n = 1$);

$n = 2$ silicon probes and $n = 1$ optic fiber implanted in the dorsal hippocampus).

After the in vivo experiments, mice were anesthetized and, in the case of the tetrode implants, an electrolytic lesion procedure was applied to mark the location of the implant tips; then mice were perfused transcardially and 50-µm coronal sections were prepared using standard histology techniques (see "Methods"). Implant tracks were examined in bright- and dark-field microscopy images based on the electrolytic lesion site and the trace of the implant along its descent. Tracks were also visualized in fluoromicrographs by red fluorescent DiI applied on the implants. Atlas sections[29] at the corresponding coronal levels were aligned with the microscopy images using linear scaling and rotation, which provided atlas coordinates of the tip and track of the implants (Fig. 7a and Supplementary Fig. 5). Histological reconstruction was then compared with the in vivo localization (Fig. 7b, c).

To go beyond this qualitative comparison and provide comparable measures of targeting accuracy, we quantified the offset relative to the target along the three cardinal axes both with in vivo localization and histological reconstruction ($n = 12$). The location of the implant tip was assessed as a range of coordinates in the anteroposterior and mediolateral directions spanned by the full extent of the radiodense area (CT) or the track in the tissue (histology), and characterized by a single dorsoventral coordinate corresponding to the most ventral point (Fig. 8a and Supplementary Figs. 6a and 7a). As expected, offset measures were strongly correlated between the in vivo and the histological localization in all dimensions (Pearson's correlation coefficient $r > 0.9$, Supplementary Fig. 8a).

To quantify the accuracy of in vivo localization, we calculated the mean absolute difference between in vivo and histology coordinates (Fig. 8b). In the dorsoventral and mediolateral directions, in vivo localization, either based on CT measurement only or relying on the CT–MRI fusion, showed an average deviation from histology of less than 0.1 mm (maximum, 0.19 with CT and 0.15 with CT–MRI fusion; no significant difference between methods or directions, $P > 0.1$ two-tailed Wilcoxon signed-rank test). In the anteroposterior direction, the CT–MRI fusion technique was significantly closer to the histological reconstruction (average, 0.05; maximum, 0.1) than the CT-based coordinate system alignment (average, 0.15; maximum, 0.29; two-tailed Wilcoxon signed-rank test, $P = 0.0024$).

Next, we compared the area overlap between in vivo and histological localization for tetrode implants ($n = 9$) in the mediolateral and anteroposterior directions, quantified as 1 minus the length of the intersecting range normalized by the length of the shorter range (overlap index, Fig. 8c). Similar to the mean difference comparison in Fig. 8b, the overlap index showed significantly better localization with CT–MRI fusion compared to CT alone in the anteroposterior direction ($P = 0.0078$), while there was no significant difference in the mediolateral direction ($P = 0.1563$, two-tailed Wilcoxon signed-rank test).

Finally, we compared the angle of the implant trajectory. The in vivo localization method provided quantitative information on the direction of the trajectory in both the coronal and sagittal planes based on the CT scans, while the histological reconstruction only provided angle measurements in the coronal plane (Fig. 8d, Supplementary Figs. 6b and 7b). Coronal offset from the target direction was strongly correlated between the in vivo and the histological localization (Pearson's correlation coefficient $r > 0.9$, Supplementary Fig. 8b), and we only detected negligible mean difference in the offsets (average, 0.92°; maximum, 3°; no significant difference from 0°, $P = 0.5364$, two-tailed $t$ test, Fig. 8e).

These results suggest that the in vivo localization method based on CT–MRI fusion has a comparable accuracy with the gold-standard histological track reconstruction. While localization

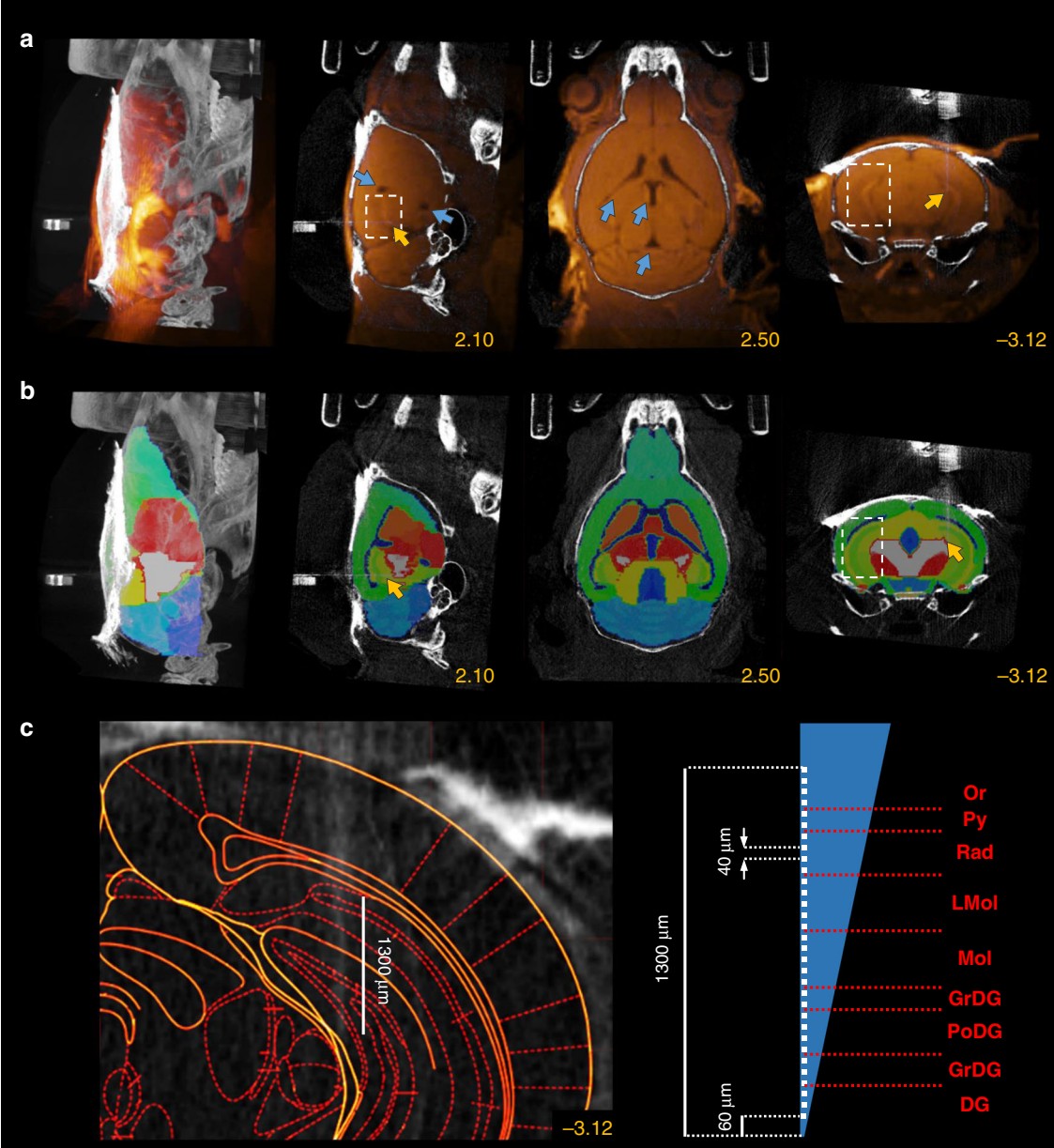

**Fig. 5 Localization of hippocampal implants with 3T MRI imaging. a** The postoperative CT image co-registered with the high signal-to-noise ratio (SNR) $T_1$-weighted preoperative MRI image shows the hippocampus (white dashed box) in addition to the ventricles (blue arrows). The yellow arrow shows the position of an edge silicon probe in the hippocampus. Images from left to right, maximum intensity projection, sagittal slice, horizontal slice, and coronal slice. Yellow numbers refer to the corresponding stereotaxic coordinates in mm relative to Bregma. **b** Three-dimensional atlas co-registered with the postoperative CT image using the CT–MRI fusion technique with special attention to the alignment of the hippocampal structure. **c** Left, localized trajectory of the probe in the hippocampus. Right, layer-specific reconstruction of 32 contact sites (Or oriens layer, Py pyramidal layer, Rad stratum radiatum, LMol lacunosum moleculare, Mol molecular layer, GrDG granular layer, PoDG polymorph layer, DG dentate gyrus). CT and MRI images are shown in grayscale and goldscale, respectively.

based on CT only provided comparable results in the medio-lateral and dorsoventral directions, it was significantly inferior to the fusion approach in the anteroposterior direction. The source of this difference might be that (i) Bregma was not unambiguously determined by the sutures, or (ii) the size of the brain was different from the size defined by the Paxinos atlas in the anteroposterior direction.

**Modes of failure revealed by the in vivo localization.** Next, we used in vivo localization to identify the exact source of targeting

error that typically remains hidden when using histology. Quantification of implantation tracks showed that our surgeons tended to implant more posterior than planned ($-0.28 \pm 0.33$ mm, mean ± standard deviation, significantly more posterior than the target, one-tailed $t$ test, $P = 0.0110$, Fig. 8f). This was due to a slight angle in the parasagittal plane ($-3.00° \pm 3.97°$, significantly more posterior than the dorsoventral direction, one-tailed $t$ test, $P = 0.0203$, Fig. 8g) that positively correlated with the targeting offset (Pearson's correlation coefficient $r = 0.74$, Supplementary Fig. 8c; the bent implant was excluded, see below; mean direction used for the dual implant). These results suggest a systematical

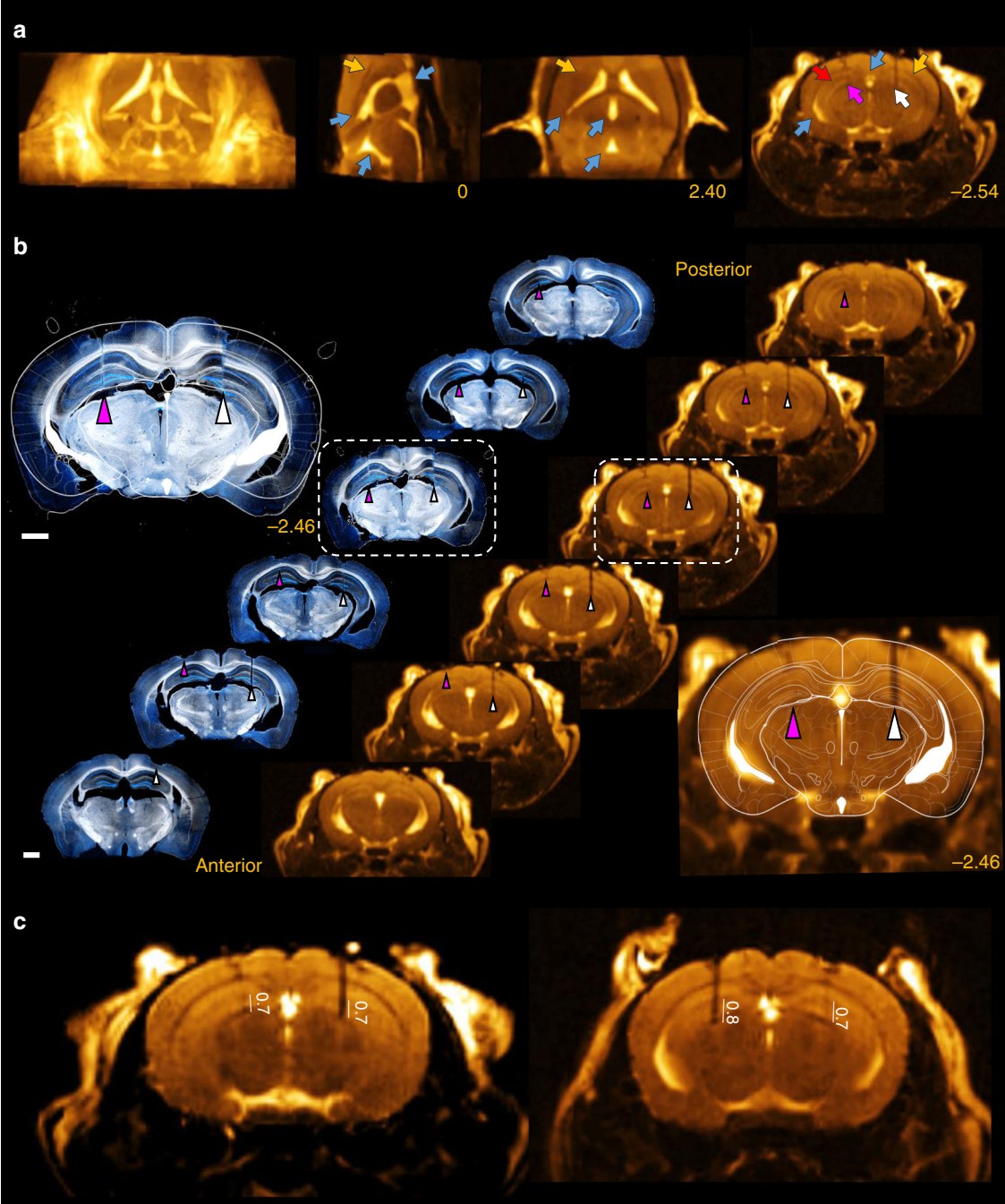

**Fig. 6 Localization of nonmetal implants with postoperative MRI imaging. a** The postoperative $T_2/T_1$-weighted MRI image shows the tracks of bilaterally implanted optic fibers (magenta arrow, 50-μm core diameter, right hemisphere; white arrow, 105-μm core diameter, left hemisphere). Anatomical landmarks are discernable, including ventricles (blue arrows), the corpus callosum (yellow arrows), and the hippocampal fissure (red arrow). Images from left to right, maximum intensity projection, sagittal slice, horizontal slice, and coronal slice. Yellow numbers refer to the corresponding stereotaxic coordinates in mm relative to Bregma. **b** Slice-by-slice comparison of the in vivo postoperative MRI images (right) and histology (left, dark-field images overlaid with fluorescent micrographs of nuclear DAPI staining). The corresponding coronal plane of the Paxinos atlas was fitted on the slices marked by white dashed boxes. Scale bar, 1 mm. **c** High signal-to-noise ratio (SNR) postoperative $T_2$-weighted MRI images were acquired from thick (600-μm) coronal slices from both animals. Numbers indicate the depth of the implant tips relative to the hippocampal fissure in mm. MRI images are shown in goldscale.

error in the stereotaxic leveling of the skull during surgery, probably owing to the ambiguity of the Lambda point that has a strong influence on anteroposterior leveling. There was no systematic deviation from the mediolateral target coordinate and from the preferred direction in the coronal plane (0.18°, mean direction used for the dual implant, Fig. 8g), although the

standard deviation was relatively large (±5.71°). As expected, the coronal angle offset was positively correlated with the mediolateral offset of the tip (Pearson's correlation coefficient $r = 0.68$, Supplementary Fig. 8c).

Sporadic errors also contributed to the relatively large standard deviations of trajectory angles. In one case, the direction of

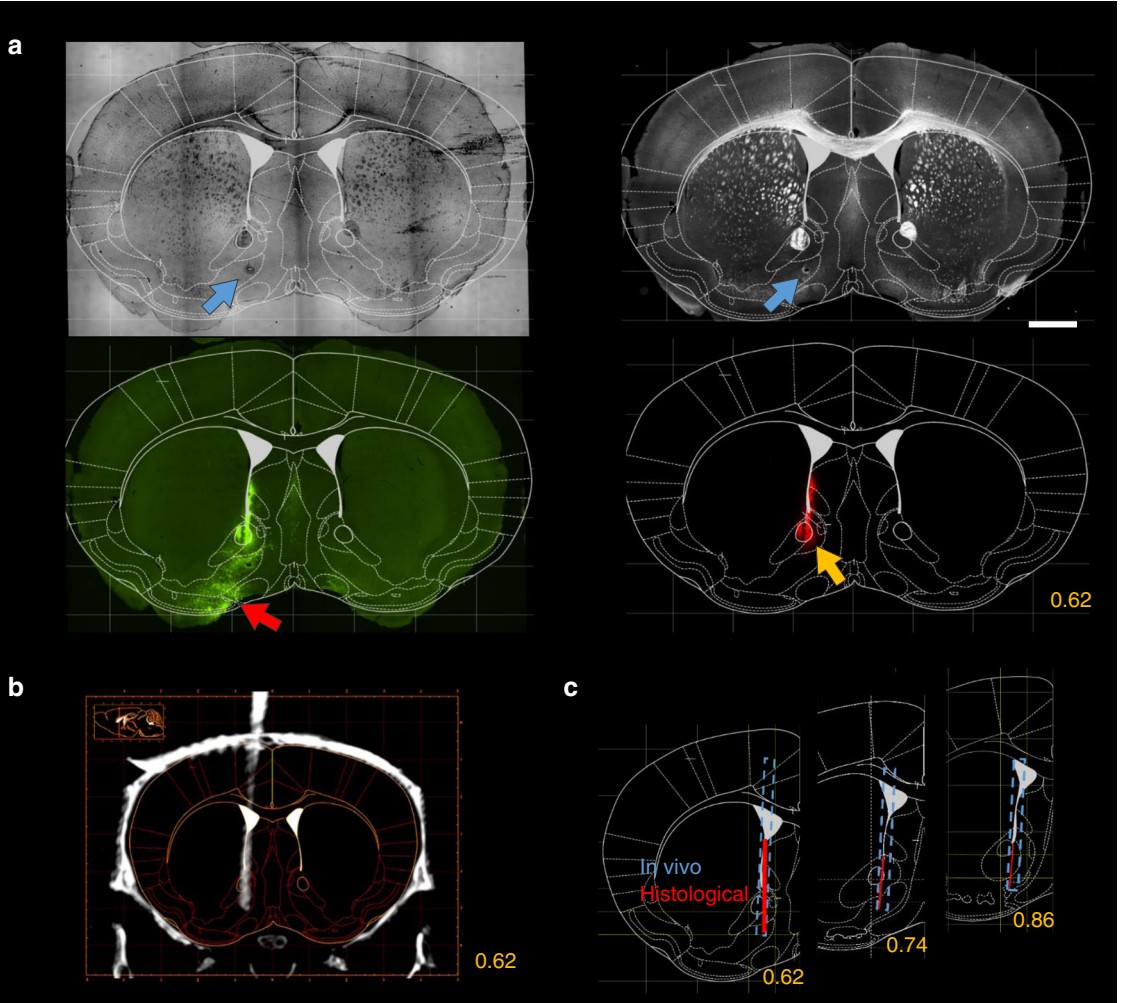

**Fig. 7 Comparison of in vivo localization and post hoc histology. a** Top, the Paxinos atlas was fitted on coronal sections based on anatomical landmarks identified in the bright-field (left) and dark-field (right) images. Blue arrows, electrolytic lesion. Bottom, the atlas image was overlaid on fluorescent images showing the virus infection at the target area (left, red arrow; green, eYFP) and the implant track (right, yellow arrow; red, DiI). Scale bar, 1 mm. **b** The corresponding coronal CT slice with the implant trajectory localized in vivo (grayscale image). **c** Comparison of the histologically reconstructed implant track (red line) and the in vivo localization (blue dashed line). Yellow numbers refer to the anteroposterior coordinates from Bregma in mm.

implantation was more than 2° anterior; the in vivo imaging revealed that this was due to bending of the tetrode bundle during the implantation procedure (Fig. 9a, blue arrow). In some surgeries, the tetrode bundle was not parallel to the guiding cannula (Fig. 9b, red dashed line), suggesting errors in the drive assembly process. We also found a damaged implant with a kink in the tetrode bundle outside the brain (Fig. 9c, yellow arrow), which suggests either a problem with the microdrive-building procedure or failing to prevent dental cement from reaching the craniotomy. In a further case, the implanted tetrode electrodes covered an unusually large area spanning multiple brain regions (Fig. 9d, blue dashed circle), indicating a larger-than-optimal collateral spread of the tetrode bundle.

**Optimization of image quality and radiation dose**. In micro-CT imaging, high resolution is achieved by decreasing the distance between the source and the object, which is inversely proportional to the square of radiation dose. In addition, a better SNR can be achieved by longer exposition time and an increased number of projections. Therefore, micro-CT imaging at high resolution and SNR may involve relatively high radiation doses; thus, possible adverse effects of radiation cannot fully be ruled out. As this issue has not hitherto been investigated in depth, we performed dose

measurements and health monitoring at a wide range of parameters to optimize micro-CT imaging with respect to this image quality–radiation dose trade-off.

We tested several acquisition settings for magnification, number of projections, and exposure time ($n = 3$ mice for each setting, Fig. 10a). Parameters that did not affect the information content of the acquired images significantly, e.g., photon energy in the available 45-kVp–65-kVp range, were set to minimize the radiation dose. The tested settings are referred to by numbers (i)–(v) as shown in Fig. 10a. Radiation dose was measured with termoluminescent detectors (TLD) attached to the neck of the animals as close to the skull as possible, where the highest dose was expected based on spatial considerations. The measured dose of a single scan ranged from 4078 to 79 mGy across settings (i)–(v). All settings were suitable for localizing all relevant bone structures. Settings (i)–(iii) visualized the position of tetrode implants with the precision required for the localization methods. For localizing silicon probes, the better SNR of settings (i) and (ii) were necessary, while the bigger-diameter optic-fiber implant could be reliably localized with the lower resolution of setting (iv) too (Supplementary Fig. 9). Most settings did not produce observable adverse health effects during the 4 weeks of the follow-up period. However, with setting (i), representing the highest

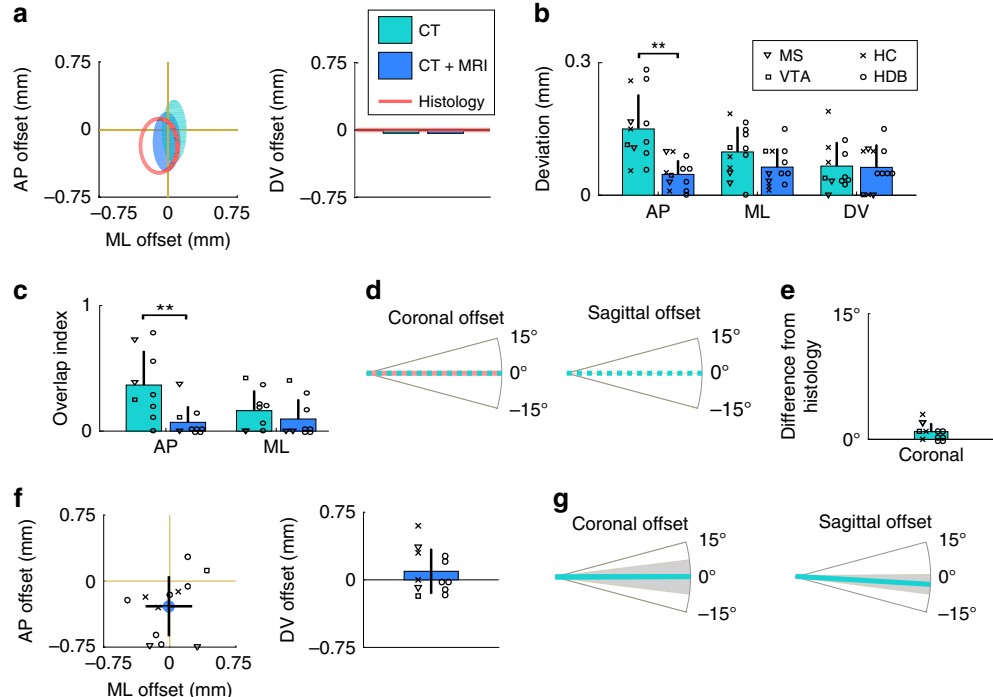

**Fig. 8 Quantification of localization accuracy. a** Anteroposterior (AP), mediolateral (ML), and dorsoventral (DV) offset from the target coordinate for an example animal. Teal, CT-based coordinate system alignment; blue, CT–MRI fusion-based atlas co-registration; salmon, histological track reconstruction. Ellipses on the left represent the area covered by the tip of the implant in the horizontal plane. **b** Mean absolute difference of tip coordinates in three directions with respect to histology. CT–MRI fusion-based localization was significantly closer to histology results in the AP direction. **P < 0.01; P = 0.0024; two-tailed Wilcoxon signed-rank test. Individual data points (n = 12 implants) are shown with different symbols representing target areas. **c** Mean distance from histology based on the overlap of implanted areas for tetrode implants. CT–MRI fusion-based localization was significantly closer to histology results in the AP direction. **P < 0.01; P = 0.0078; two-tailed Wilcoxon signed-rank test. Individual data points (n = 9 implants) are shown with different symbols representing target areas as in panel **b**. **d** Coronal and sagittal offset from the planned direction of the trajectory for the same example animal as in panel **a**. Implant directions were quantified based on the CT images. Histology only provides coronal offset measures. **e** Mean absolute difference of CT-based coronal angle compared to histology. Individual data points (n = 12 implants) are shown with different symbols representing target areas as in panel **b**. **f** Mean offset from the target of the centroid of the localized areas with CT–MRI fusion-based localization. Individual data points (n = 12 implants) are shown with different symbols representing target areas as in panel **b**. **g** Mean offset from the coronal (n = 12 implants) and sagittal (n = 11 implants) target direction with CT-based localization. Anterior, lateral, and ventral directions were defined as the positive directions. Error bars and gray shadings in panel **g** show mean ± standard deviation.

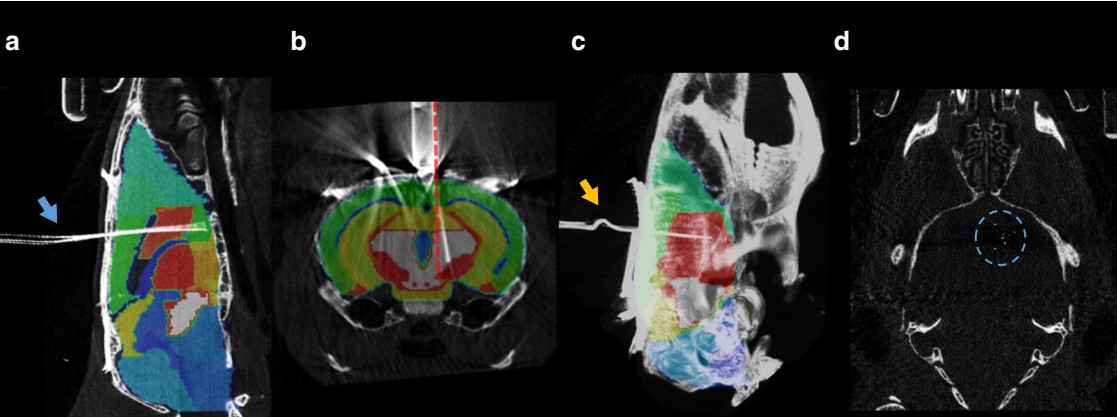

**Fig. 9 Modes of failure revealed by the in vivo localization. a** Bending of the tetrode bundle (blue arrow). **b** Tetrode bundle not parallel with the cannula (red dashed line, the cannula was aligned with the dorsoventral direction during the stereotaxic surgery). **c** Kink on the tetrodes outside the brain (yellow arrow). **d** Overdispersion of the tetrodes, covering multiple adjacent brain areas (blue dashed circle). CT images are shown in grayscale.

radiation dose, we observed a slight loss of hair and skin irritation around the neck of the animals after 2–3 weeks. Therefore, we restricted further scans to settings (ii)–(v).

Next, we subjected another cohort of animals (n = 3 mice for each group) to the full protocol of two micro-CT scans separated by 1 week, and monitored the weight and general condition of the

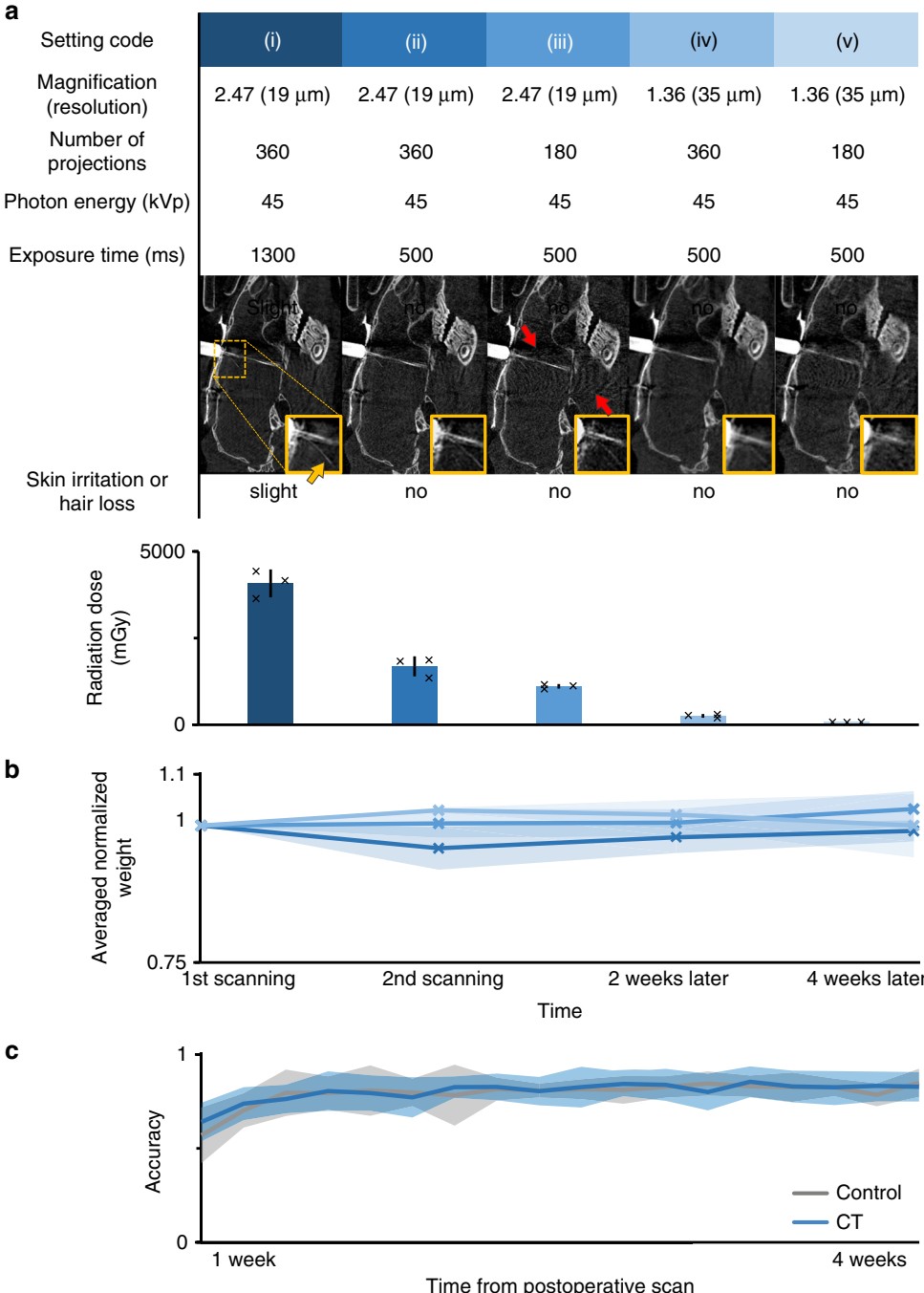

**Fig. 10 Optimization of image quality and radiation dose. a** Top, acquisition settings for CT; middle, corresponding image of a tetrode drive implant (grayscale images; for silicon probe and optical fiber implants, see Supplementary Fig. 9); bottom, radiation effects and measured doses ($n = 3$ mice in each group; $n = 15$ mice in total). Error bars show standard deviation from the mean; markers show individual data points. The yellow arrow shows a bent single tetrode in the magnified image, indicating the spatial resolution of the images. Red arrows show artifacts caused by the metal parts of the microdrive, more apparent for lower exposure time and number of projections. **b** Average normalized body weight of mice ($n = 3$ mice in each group, $n = 9$ mice in total) exposed to two CT scans separated by a 1-week pause, mimicking the protocol used for the localization procedure. The first scan was performed with setting (v), and the settings for the second scan are indicated by the color code adopted from panel **a**. Error shades show standard deviation from the mean. **c** Performance of scanned animals ($n = 7$ mice, settings (v) and (iii), blue) in a standard five-choice serial reaction time task did not differ (one-tailed Mann–Whitney $U$ test for each day, $P > 0.1$) from that of the control population ($n = 7$ mice, gray). Error shades show mean ± standard deviation.

animals. In all groups, the first scan was performed with setting (v) characterized by the lowest, negligible dose that was still suitable for localizing the relevant bone structures. The second scan was performed with settings (ii)–(iv). None of the animals showed any signs of skin irritation, hair loss, cataract[34,35], or drop in weight (Fig. 10b). Based on these findings, we suggest that postoperative CT parameters should be adjusted according to the size and radiodensity of the implant, with the lowest possible radiation dose.

Finally, we compared the performance of mice ($n = 7$) scanned with setting (v) before and setting (iii) after surgery with a control population ($n = 7$) not exposed to radiation in a standard

five-choice serial reaction time task[36]. Before the surgery, the animals were pretrained for 5–10 days in an automated system[37] (see "Methods"). By the end of the pretraining period, all animals learned the task and performed at around 80% accuracy. One week after the surgery, postoperative scanning was performed, and the training was resumed. Those mice that had been subjected to the scanning procedure performed at a level of accuracy indistinguishable from the control population (Fig. 10c, the scanned population did not perform significantly worse on any of the training days, one-tailed Mann–Whitney U test, P > 0.1).

## Discussion

Good temporal resolution and direct connection with neural activity make electrophysiology and optogenetics powerful techniques to study or manipulate neural activity of awake-behaving animals. Developed recently, fiber photometry experiments provide a cell-type-specific readout of the ongoing activity of neural populations. When complicated implants and difficult surgeries, often followed by complex and lengthy behavioral training, are increasingly commonplace, variability in targeting accuracy of stereotaxic surgeries is becoming rate-limiting. Consequently, the lack of in vivo methods for localizing implanted electrodes or optic fibers eventually thwarts the effectiveness of these studies.

Structural imaging has been used to localize implants in humans, monkeys, and rats. For instance, Borg et al.[26] used micro-CT imaging in rats for localizing 50-μm-diameter microwire electrode arrays postmortem with an accuracy sufficient for the larger structures of the rat brain (with at least 1000-μm width and height). Also in rats, Rangarajan et al.[27] localized 200-μm-diameter electrodes and lesions in vivo with CT and MRI, respectively. However, in vivo localization of thin electrodes or optic fibers implanted into small anatomical structures of the mouse brain remained a challenge, and therefore in vivo implant localization has not been performed in mice.

To resolve this issue, we developed a noninvasive in vivo localization technique for radiodense implants in mice by combining different structural imaging modalities. We were able to localize deep-brain implants targeting small nuclei as well as superficial implants targeting layered structures, with a precision comparable to postmortem histological reconstruction. The method largely relies on transforming the co-registered pre- and postoperative micro-CT images to the stereotaxic reference system, capable of determining the coordinates of the implant relative to Bregma (Figs. 2 and 3). This method was aided by a preoperative MRI image, which was co-registered with both a three-dimensional MRI-based atlas and the postoperative CT image in a multistep protocol based on anatomical landmarks (Fig. 4). The added benefit of the structural MRI was that it took individual anatomical variation into account and, by introducing soft-tissue contrast absent in the CT modality, provided visual three-dimensional information about the trajectory of the implant in the brain. While the CT-based alignment was sufficiently precise to localize small-diameter implants in deep-brain nuclei with high accuracy, localization based on the CT–MRI fusion was significantly more accurate along the anteroposterior axis in the sagittal plane. Although we did not find a significant difference between the CT- and fusion-based methods in the mediolateral direction, we note that implants farther from the midline could possibly reveal a larger difference.

We further showed that nonmetal implants can be localized by postoperative MRI scans, otherwise prevented by metal objects in the case of electrodes (Fig. 6). In such applications, localization precision is determined by image resolution. We used this approach to image fiber-optic implants useful for optogenetics and fiber photometry experiments; however, MRI-compatible silicon probes are becoming commercially available in a rapidly growing number, foreshadowing a wider applicability of postoperative MRI-based in vivo localization.

Verifying the success of implantation directly after surgery using in vivo, noninvasive techniques allows precise adjustment of the recording depth with microdrives, increasing targeting precision and thus experiment efficiency. Moreover, if the implant proved to have missed the target location, which may occur in up to 30–60% of surgeries for deep-brain targets according to previous[27,38] and our estimates, early termination saves valuable time, human, and other lab resources. For instance, in a previous study[39], we implanted 32 mice to target the basal forebrain for optogenetic tagging of central cholinergic neurons. Optogenetic effects detected in 26/32 mice indicated that the electrodes and optic fiber approached cholinergic areas, prompting us to perform full histological reconstruction. Post hoc localization determined that the basal forebrain was hit in 19/32 animals. In the remaining 13 mice, 374 recording sessions were performed, each lasting approximately 2 h that included behavioral training and two optogenetic tagging sessions. Thus, we estimate that early termination based on in vivo localization would have saved about 750 net working hours throughout the course of the project, while shortening descent time in the successfully implanted mice could have further optimized experimental time. Indeed, we often adjust dorsoventral position of microdrive implants following in vivo localization. Behavioral experiments especially benefit from the fact that uncertainty about the distance between the implant and target zone is this way lifted, allowing proper temporal coordination of animal training and recording.

We showed that in vivo localization based on the CT–MRI fusion technique, and postoperative MRI of metal-free implants, provides the same level of accuracy as histological track reconstruction (Figs. 6 and 8) with comparable time investment (~2–3 h per animal with CT–MRI fusion, and 1 h per animal with postoperative MRI scans). In addition, the in vivo method is less subjective and more reliable in the sense that it is free from the irreversible information loss that may occur during slice preparation—for instance, tissue shrinkage during perfusion, damage caused by removing the implant, suboptimal angle of slice preparation, and tissue damage or loss during sectioning, which can be especially problematic when multiple implants are positioned close to each other. We suggest that in projects using in vivo localization, it becomes unnecessary to sacrifice the animal immediately after the experiment. This opens the possibility of designing long-term experiments, where the animal may be followed up after the training process, tested for long-term memory, or trained on different tasks.

Targeting accuracy depends on a number of factors including scaling errors when using animals of different age, size, or strain, errors in aligning to the axes of the atlas system, or subtle displacements of the skull due to drilling. Another benefit of visual feedback on surgical procedures is the chance of identifying potential causes of failure, including systematic errors committed by the operators that often remain hidden when using histology. Indeed, we discovered a tendency of implanting posterior from the target position due to deviation from the vertical trajectory in the sagittal plane that suggested a slight error in Bregma–Lambda leveling. Other causes of mistargeting included bending or overspreading of the tetrode bundle and damaged implants that suggested problems with microdrive building or surgical procedures (Fig. 9). These procedural errors are likely not unique to our operators, providing general cautionary notes for all mouse surgeons on a few issues that might benefit from extra attention. For instance, anteroposterior leveling suffers from uncertainties

about the Lambda point, detrimental due to the level difference between the occipital and parietal plates, potentially amplified by the smaller skull size of mice compared to rats. This typically results in a tilt of the head causing deviation from the planned trajectory in the posterior direction, which could be systematically compensated during the leveling procedure. Our findings also demonstrated the underappreciated importance of parallel positioning of guiding tubes, individual electrodes, and optic fibers with the stereotaxic arm.

We showed that the structural information provided by the MRI imaging improved the accuracy of the in vivo localization along the anteroposterior axis (Fig. 8b). This suggests that sub-optimal surgical procedures are not the exclusive source of targeting inaccuracy, but precision is further limited by the individual anatomical variability of the target coordinates, as well as deviations of the bone-defined Bregma point from the idealized origin of the atlas coordinate system[40–42]. Applying CT–MRI fusions, one might adjust Bregma position or optimize target coordinates for each animal individually, as implemented in human surgical planning and in some monkey implantation surgeries[22,23,43,44]. Thus, preoperative imaging for surgical planning can further save time and resources, especially in studies where animals have to be pretrained on a behavioral task before the surgery or the availability of subject animals is limited.

Besides tetrode implants, chronic silicon probe and fiber-optics applications may also benefit from in vivo localization for several reasons. (i) Optogenetic manipulations are often targeted at neuron types that are not restricted to the target zone (e.g., GABAergic, glutamatergic, and parvalbumin-expressing), in which case precise positioning of the optic fibers becomes important, (ii) large, stiff optic fibers may introduce lesions in the target zone if implanted too deep, compromising the interpretation of the experiment, (iii) fiber positioning in fiber photometry experiments is important in order to gain a high SNR signal from the targeted population, (iv) silicon probes often bend on white matter tracts, complicating precise aiming, and (v) localization of silicon probe tips is difficult with histology; since CT images visualize the probe from the base, the known length of the probe provides verification of the tip location.

Since CT-based in vivo implant localization has not yet been performed in mice, we investigated the incurred radiation dose to provide recommendations for parameter settings that minimize radiation without compromising localization accuracy. We optimized our scanning procedures to provide sufficient quality without adverse effects on health, including weight, integument, eye lenses[34,35], and performance in behavioral tasks. We provided recommendations for acquisition settings for pre- and post-operative micro-CT scans based on the ALARA principle ("as low as reasonably achievable") for human radiation protection[45] (Fig. 10).

Noninvasive, in vivo imaging modalities are essential to the longitudinal study of animal models in many biomedical research areas, including oncology, neurology, cardiology, and drug research[46–50]. We believe that the in vivo localization method presented here might become a similarly important tool in longitudinal neurophysiology experiments. With the widespread use of small animal imaging techniques in most fields of biomedical sciences[51], preclinical CT and MRI equipment are readily accessible at many universities and research institutes. In addition, functional MRI measurements[52–55] have already facilitated the installation of MRI machines, capable of high-precision implant localization, at a large number of neuroscience research centers. Nevertheless, we showed that either micro-CT or MRI imaging of metal-free implants might be sufficiently precise for part of the applications.

## Methods

**Animal care and use**. Wild-type ($n = 24$) and genetically modified (ChAT-Cre, $n = 22$; ChAT-Cre × DAT-Cre, $n = 4$, DAT-Cre, $n = 5$) adult (3–4 months old, 25–30 g of weight) male mice of C57BL/6N genetical background were used. Animals were housed individually in $36 × 20 × 15$-cm cages under a standard 12-h light–dark cycle (lights on at 8 a.m.) with food and water available ad libitum. Temperature and humidity were kept at $21 ± 1$ °C and 50–60%, respectively. All experiments were approved by the Institutional Animal Care and Use Committee and the Committee for Scientific Ethics of Animal Research of the National Food Chain Safety Office (PE/EA/675-4/2016, PE/EA/1212-5/2017, and PE/EA/864-7/2019), and were performed according to the guidelines of the institutional ethical code and the Hungarian Act of Animal Care and Experimentation (1998, XXVIII, section 243/1998, renewed in 40/2013) in accordance with the European Directive 86/609/CEE and modified according to the Directives 2010/63/EU. B.K., I.H., K.Sz., and B.H. each hold a certificate course diploma of Advanced Radiation Protection (with certification numbers H04/2017, B-2019/54, OSSKI-2014-ÁK-369-21, SUVE-B-059/2008), granted by the Budapest University of Technology and Economics or the Semmelweis University, Budapest, according to the Hungarian Act CXVI of 1996 on Atomic Energy ("Atomic Act"), granted based on the permission of the National Public Health Service of Hungary.

We followed the ALARA principle of human radiation protection during the animal scans. The ALARA principle states that all use of ionizing radiation should be optimized so that our goal is reached by the lowest reasonably achievable dose and risk. Specifically, to prevent unnecessary exposure and overexposure of the animals, we applied the following. First, we minimized radiation dose to the lowest reasonably achievable level by the three basic tools of radiation protection: time, distance, and shielding. Exposure time was minimized by optimizing scanning parameters (Fig. 10). Distance was optimized by performing preoperative CT scans at lower resolution (larger distance) compared to the higher resolution required for postoperative scans (see "Optimization of image quality and radiation dose"). Animals not being scanned were kept in a room shielded from the CT. Second, radiation dose was minimized by using the lowest available X-ray photon energy (see "Optimization of image quality and radiation dose"). Third, we minimized unnecessary exposure by using a single X-ray image (with negligible dose) to precisely and reliably set a minimal scan area (see "Preoperative scanning" section). Fourth, we waited for at least 1 week between two CT scans to exclude the risk of adverse health effects.

**Stereotaxic surgery**. Animals were anesthetized using an intraperitoneal injection of 25 mg kg$^{-1}$ xylazine and 125 mg kg$^{-1}$ ketamine in 0.9% NaCl. The skin and subcutaneous tissues of the scalp were infused by Lidocaine to achieve local anesthesia, and the eyes were protected with ophthalmic lubricant (Corneregel eye gel, Bausch & Lomb, Rochester, NY, USA). The mouse was placed in a stereotaxic frame (David Kopf Instruments, Tujunga, CA, USA). The skin was incised, the skull was cleaned and leveled, and a cranial window was opened above the target area (HDB, + 0.74 mm anteroposterior, 0.6 mm lateral; MS, + 0.9 mm anteroposterior, 0.9 mm lateral; VTA, −3.1 mm anteroposterior, 0.6 mm lateral; ventral CA3, −2.5 mm anteroposterior, 2 mm lateral; dorsal CA1 at different locations, −2 mm anteroposterior, 1.5 mm lateral; −2.5 mm anteroposterior, 2 mm lateral; −3 mm anteroposterior, 2 mm lateral). Injections of the adeno-associated virus vector encoding ChR2 (AAV2.5.EF1a.Dio.hChR2(H134R)eYFP.WPRE.hGH, 300 nl, titer: $7.7 × 10^{12}$ GC ml$^{-1}$) (Penn Vector Core, PA, United States) were performed using glass pipettes pulled from borosilicate capillaries broke to 20–30-μm tip diameter, connected to a MicroSyringe Pump Controller (World Precision Instruments, Sarasota, FL, USA), lowered stereotaxically into the target nucleus (HDB, DV −5.0/−4.7 mm; MS, DV, −3.5/−4.2/−4.6 mm; VTA, DV −4.4/−4.0 mm, while no virus injection was performed in the case of the hippocampal targets).

Custom-built microdrives[39] housing eight nichrome tetrodes (diameter, 12.7 μm, Sandvik, Sandviken, Sweden) and a 50-μm core optic fiber (outer diameter, 65 ± 2 μm, Laser Components GmbH, Olching, Germany) were implanted into the target nuclei with a metal-free guiding cannula that did not enter brain tissue, using stereotaxic positioning to maintain the dorsoventral orientation of the implant. Tetrode bundles covered an approximately cylindrical volume in the tissue with nonuniform gaps between the tetrodes, with an average diameter of 0.34 ± 0.07 mm based on the CT images. Optic fibers (core diameter, 50 μm; outer diameter, 65 ± 2 μm, Laser Components GmbH, Olching, Germany or core diameter, 105 μm; outer diameter, 250 ± 4 μm, Thorlabs Corp., Munich, Germany) or silicon probes (Buzsaki-type, Buzsaki16 with a single 52 × 15-μm shank; Neuronexus Technologies, Ann Arbor, MI, USA; polytrode probe, A1x32-Poly2 with a 113 × 15-μm shank, Neuronexus Technologies, Ann Arbor, MI, USA; custom-built edge probe with a 150 × 15-μm shank, Neuromicrosystems, Budapest, Hungary) were implanted at a fixed depth in the target area (see Supplementary Tables 1 and 2 for implant dimensions and implantation coordinates, respectively).

Fluorescent dye (DiI, Invitrogen LSV22885, Ottawa, Canada) was applied on the tip of the tetrode and silicon probe implants before lowering for later histological localization. Implants were secured by dentil adhesive (C&B Metabond, Parkell, Edgewood, NY, USA) and acrylic resin (Jet Denture, Lang Dental, Wheeling, USA). Buprenorphine was used for postoperative analgesia (Bupaq, 0.3 mg ml$^{-1}$, Richter Pharma AG, Wels, Austria).

**Behavioral training**. Mice ($n = 14$) were trained for 30 min daily in a custom-built setup[56] on a five-choice serial reaction time task[36]. Mice were water-restricted to ~85–90% body weight and received water reward after correct trials. Each trial started with a 3–5-s-long intertrial interval (ITI) when poking was not allowed. Next, one of five ports was illuminated, and the animal had to detect this port by a nose poke during the illumination or a short time period after that, to receive 4–6 µl of water reward from the same port (correct choice). The length of the illumination was shortened throughout the training stages[36]. Poking during the ITI, choosing an incorrect port, or omitting a response was punished by a 5-s-long timeout, during which the house light was turned off and no new trial was started. All mice were pretrained before the surgery in a custom-built automated training system[37], where they were allowed to enter the training setup from their home cages and perform the task regularly every 2 h for 15 min (12 times a day), until they reached 80% performance (5–10 days). After 1 week of recovery period following the surgery, postoperative CT scans were performed in the case of the test group ($n = 7$, allocated randomly), and the behavioral training was resumed. Behavioral performance was assessed by choice accuracy, defined as the ratio of the correct and the total number of choices (correct and incorrect)[36].

**Atlases**. The Bregma and Lambda points and the axes of the stereotaxic coordinate system were defined according to Paxinos and Franklin. Bregma and Lambda were determined as "the midpoints of the curve of best fit along the coronal and the lambdoid suture, respectively. They are not necessarily the points of intersection of these sutures with the midline suture"[29]. The same atlas was used for surgical planning and comparing coordinates across localization methods. For localization based on CT–MRI fusion, a three-dimensional mouse brain atlas by Bai et al.[32] was used, which is based on the $T_2$-weighted MRI images of five mice with structures manually identified on the basis of the Paxinos atlas.

**Preoperative scanning**. Preoperative CT and MRI scanning was performed in the same scanning bed to avoid postural changes and disparities between the imaging planes. Isoflurane gas was used as an inhalation anesthetic through a specialized isoflurane mask that was also used for head fixation to avoid motion artifacts (similar to ref. [28]). Scanning was controlled using the Nucline 2.01 (Mediso Medical Imaging Systems, Budapest, Hungary) software.

CT measurements were performed on a NanoX-CT (Mediso Medical Imaging Systems, Budapest, Hungary) cone-beam micro-CT imaging system with an 8-W power X-ray source. Following the ALARA principle, to minimize radiation dose, we used simple circular scanning to scan the head of the animal only. To this end, the head of the animal was positioned at the same marked area on the scanning bed. We took a single X-ray image of this area with a negligible radiation dose of less than 5 mGy and used this image to set the final scan area. Preoperative CT imaging parameters were set to minimize the radiation dose: tube voltage, 45 kVp; magnification, 1.36; exposure time, 500 ms; number of projections, 180. These represent minimal values within the adjustable range with 3-min acquisition time.

Magnetic resonance imaging was performed with a nanoScan PET/MRI system (Mediso Medical Imaging Systems, Budapest, Hungary), which is equipped with a permanent magnetic field of 1 T and with a 450 mT m$^{-1}$ gradient system using a volume coil for both reception and transmission. Fast $T_1$-weighted images were acquired with a 3D gradient echo sequence using 8 excitations, $T_R = 15$-ms repetition, and $T_E = 2.2$-ms echo times and 25° flip angle with resolution set to 0.28 mm. The sequence parameters were selected in order to achieve proper contrast and SNR that facilitates good visualization of the contour of the brain and the position of the ventricles with sufficient resolution in reasonable acquisition time (10 min).

Translational-field magnetic resonance imaging was performed with a nanoScan 3T MRI system (Mediso Medical Imaging Systems, Budapest, Hungary) with a 3T cryogen-free superconducting magnet and with a 600 mT m$^{-1}$ gradient system using a 72-mm volume coil for transmission and a 20-mm surface coil for signal reception. High SNR $T_1$-weighted images were acquired with a 3D gradient echo sequence using three excitations, $T_R = 25$-ms repetition and $T_E = 2.7$-ms echo times and 30° flip angle with resolution set to 0.2 mm, which enabled us to visualize the hippocampal formation with 20-min acquisition time.

**Postoperative scanning**. Postoperative CT scans were performed using a similar procedure as detailed above, 4–12 days after the surgery. In case the implant was lowered during the in vivo electrophysiology recordings, the postoperative CT scanning was repeated after the termination of the experiments for accurate comparison of in vivo localization with the histology results (see "Quantification of localization accuracy using gold-standard histology" section). Metal parts of the drive were positioned to point away from the plane in which the X-ray source was rotating, so their shadow artifacts bypassed our region of interest, i.e., the close environment of the implant. Acquisition parameters were set to achieve sufficient image quality at the minimal possible radiation dose in accordance with the ALARA principle. Postoperative acquisition settings were tested in the range of 500–1300-ms exposure time, 180–360 projections, and 1.36–2.47 magnification, with the tube voltage set to 45 kVp. Optimal acquisition settings, depending on the size and radiodensity of the implant, are provided in the "Optimization of image

quality and radiation dose" section of "Results". In the presence of metal parts, no MRI scanning was performed after the surgery.

In the case of the two mice implanted with bilateral optic fibers in the hippocampus, we performed postoperative MRI imaging with the 3T system, 1 week after the surgery. High-resolution $T_2/T_1$-weighted 3D-balanced steady-state free-precession (bSSFP) sequence was performed using 12 excitations, $T_R = 5.13$-ms repetition, and $T_E = 2.565$-ms echo times and 40° flip angle with resolution set to $0.12 \times 0.12 \times 0.15$ mm. High-resolution $T_2$-weighted 2D fast spin echo sequence was performed using 20 excitations, $T_R = 3000$-ms repetition, and $T_E = 98.9$-ms echo times with resolution set to $0.1 \times 0.1 \times 0.6$ mm. The sequence parameters were set to achieve the best resolution and image quality, suitable for visualizing the track of both implants in the hippocampal formation, in reasonable acquisition time (20 min).

**Image reconstruction and registration**. CT image reconstruction was performed with the Nucline 2.01 (Mediso Medical Imaging Systems, Budapest, Hungary) software, using filtered back projection with a Butterworth filter[57]. The isotropic voxel size was set to the minimum value of 35 µm or 19 µm for 1.36 and 2.47 magnifications, respectively. Coordinate system transformations and CT–MRI-atlas co-registrations were performed with the VivoQuant 1.22 (inviCRO, Boston, MA, USA) preclinical medical image post-processing software, using Euclidean transformations (translation and rotations, except for step (iii) of the CT–MRI fusion method, where non-Euclidean affine transformations were used), based on anatomical landmarks. Electrode tracks were segmented with an intensity threshold applied on a whole-brain ROI.

**Histological track reconstruction**. After termination of the experiments, mice were deeply anesthetized with ketamine–xylazine (see above). In animals implanted with tetrode drives, to accurately mark the position of the electrode tip for histological reconstruction, electrical lesioning (40 µA for 5 s applied through one or two leads; IonFlow Bipolar, Supertech Instruments, Pécs, Hungary) was performed. Mice were then perfused transcardially with 0.1 M phosphate-buffered saline (PBS) for 1 min, then with 4% paraformaldehyde (PFA) in PBS for 20 min. Implants were carefully removed, brains were extracted from the skull and postfixed for 24 h in PFA at +4 °C.

Fifty-µm-thick coronal sections were cut (Vibratome VT1200S, Leica, Wetzlar, Germany). Slices from the two animals with postoperative MRI scans were stained with a 10 mg ml$^{-1}$ 4′-6-diamidino-2-phenylindole dye (DAPI, Dihydrochloride, Merck Millipore, Burlington, MA, USA). Slices were mounted on microscope slides, covered in mounting medium (Vectashield, Vector Labs, Burlingame, CA, USA), and examined with a fluorescent microscope (Nikon Eclipse Ni microscope, Nikon Instruments, Melville, NY, USA). Dark-field, bright-field, and fluorescent images were taken with a Nikon DS-Fi3 camera. Images were processed to reconstruct the track of the implants. Mouse brain atlas sections[29] were co-registered with the corresponding coronal slices using linear scaling and rotation based on anatomical landmarks identified on the bright-field and dark-field images. The atlas images were carried over to fluoromicrographs that visualized the infected area by virally transfected ChR2-eYFP (deep-brain implants) or nuclear DAPI staining (hippocampal implants) and the trajectory of the implant by fluorescent red DiI applied on the implants. The atlas fits were confirmed by the presence of virally transfected neurons in the target areas. Implant tip coordinates and trajectory directions were determined based on the DiI marked tracks and the electrical lesions localized in the bright-field and dark-field images.

**Radiation dose measurements**. Calibrated lithium fluoride MCP-N (LiF:Mg, Cu, and P) thermoluminescent dosimeters (TLD) were used to measure radiation exposure. These types of TLD chips have a broad linear dose range[58], from cGy-s to ~10 Gy. Before CT scanning, TLD detectors were attached to the neck of the animals as close to the skull as possible. After the irradiation, radiation doses were read from the detectors with a TLD Cube (RadPro, Wermelskirchen, Germany) reader system.

**Data analysis**. Analyses were carried out in VivoQuant 1.22 (inviCRO, Boston, MA, USA) and Matlab R2016a (Mathworks, Natick, MA, USA) using built-in computational, statistical, and plotting functions.

**Statistics and reproducibility**. Statistical analyses were carried out in Matlab R2016a (Mathworks, Natick, MA, USA). No statistical methods were used to predetermine sample size. However, sample size was chosen based on other studies of the field, most importantly ref. [27]. The localization procedure combining CT and MRI imaging was performed independently for $n = 12$ implantations involving three different implant types (tetrodes, silicon probes, and optic fibers) and targeting four different brain areas (HDB, VTA, MS, and hippocampus). In addition to this, independent CT-based localization was performed for two optic fibers, and a silicon probe implant and two optic-fiber implants were localized based on postoperative MRI scans. Implants could be reproducibly localized in all of these experiments, as demonstrated in the figures. For the behavioral test, 7 out of 14 animals were randomly allocated into the test group. Behavioral trials were presented in randomized order. Other parts of this study did not involve separate

experimental groups. Experimenters were not blinded to group allocation. Pre-operative pretraining was performed in a fully automated system without any interaction with the experimenters[56]. After the surgery, training was performed in a semiautomated system with minimal interactions, limited to transferring the animal from the home cage to the training chamber. Analysis of behavioral performance was fully automated. One animal with a bent implant (Fig. 9a) was excluded from a subset of analyses (noted in the text, Fig. 8f–g and Supplementary Fig. 8c) because it would have distorted the estimations of systematical implantation errors during the surgical procedure. Correlations were tested using Pearson's correlation coefficient (r), at a significance level of 0.05. The error measures were tested for statistical significance by the nonparametric two-tailed Wilcoxon signed-rank test. Systematic deviations from the target coordinate and directions were tested with one-tailed $t$ test, since approximately normal distribution could be reasonably assumed (normality was tested with and not rejected by Lilliefors test, $P > 0.05$) and the null hypothesis was one-sided for these tests. Behavioral differences between the scanned and the control group were tested by the non-parametric Mann–Whitney $U$ test using one-sided null hypothesis.

**Reporting summary**. Further information on research design is available in the Nature Research Reporting Summary linked to this article.

## Data availability

Co-registered micro-CT and MRI images were made available via Figshare (https://doi.org/10.6084/m9.figshare.12700997.v3). Source data are provided with this paper. Specifically, source data underlying Figs. 8 and 10 and Supplementary Figs. 6, 7, and 8 are provided as source data files.

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

## Acknowledgements

We thank Gabriella Taba for the TLD detectors and help with the radiation dose measurements; Dr. Andor Domonkos for providing a Buzsaki-type silicon probe; Drs. Sergio Martínez-Bellver, Viktor Varga, and Duda Kvitsiani for helpful comments and discussions on the paper. We acknowledge the help of László Barna and the Nikon Center of Excellence at the Institute of Experimental Medicine, Nikon Europe, Nikon Austria, and Auro-Science Consulting for kindly providing microscopy support. This work was supported by the "Lendület" Program of the Hungarian Academy of Sciences (LP2015-2/2015), NKFIH KH125294, and the European Research Council Starting Grant no. 715043 to B.H., the ÚNKP-19-3 New National Excellence Program of the Ministry for Innovation and Technology to B.K., and the Thematic Excellence Programme (TKP) of the Ministry of Innovation and Technology of Hungary, within the framework of the BIOImaging Excellence Programme at Semmelweis University to K.Sz. and D.M. B.H. was a member of the FENS-Kavli Network of Excellence.

## Author contributions

The idea was developed by B.H. and B.K. B.K. and I.H. designed the scanning protocol. B.K., I.H., M.B., and G.B. performed the in vivo imaging experiments. B.K. developed the in vivo localization protocol, performed the radiation dose measurements, analyzed the data, and prepared the figures; D.B., N.S., K.S., and B.K. performed the implantation surgeries; D.B., K.L., and B.K. performed histological localization; D.B. and E.B. performed the behavioral tests. The project was supervised by B.H., K.Sz., and D.M. This paper was written by B.K., B.H., and D.B. with comments from all authors.

## Competing interests

D.M. is a shareholder and employee of CROmed Ltd. Activities of the company do not interfere with the subject of this publication. B.G. and B.M. are both employees of Mediso Medical Imaging Systems. Activities of the company do not interfere with the subject of this publication. The remaining authors declare no competing interests.
