## [Peer Review File · Nature Communications]

Reviewers' Comments:

Reviewer #1:

Remarks to the Author:

Kiraly et al presented a new platform to localize deep brain implants in mice. This platform is based on micro-CT and MRI, which respectively provides bone structural information and soft tissue contrast information. By registering pre-surgical micro-CT and MRI, and align the images to post-surgical micro-CT including the implants, the authors report location precision comparable to histological reconstruction. The obvious benefit is that CT/MRI based location is non-invasive and can be conducted repeatedly. Moreover, the localization information can be used to guide the continuation or early termination of experiments, and substantially increase the efficiency of experiments involving deep brain implants such as electrodes or optic fibers.

Using this new localization platform, the authors also identified systematic targeting errors, especially in the AP (antero-posterior) direction. This is likely due to insufficient leveling between bregma and lambda, leading to a slight tilt of the skull, and hence introduces an unwanted angle of the implant in the AP direction. This problem is amplified in mice because the size of the skull is smaller, and thus the same the leveling error would lead to a bigger angular error (compared to the same error made in a larger animal such as the rat).

The authors also characterized the spatial resolution of micro-CT under different imaging parameters and the corresponding radiation exposure dose, and demonstrated that such exposure doses were safe and did not adversely affect animals' behavior.

Overall, I find the study to be meticulously conducted and clearly reasoned and written. The results are sound. The idea is not original (for example Borg et al 2015 cited in this manuscript), but the current implementation is substantially better and provides conceptual insights about reasons for targeting error, as well as recommended parameters for using this method.

A minor concern though is that the results are relevant to only a small group of researchers who target deep brain structures where the localization errors are more significant. And perhaps the most relevant audience are those who implant moveable tetrode (which are much smaller than optic fiber implants) that need to travel a long distance before reaching the target.

Also, the detailed design of the specialized isoflurane mask that maintains head fixation during micro-CT/MRI imaging should be made available to other scientists to aid the dissemination and replication of this method.

Reviewer #2:

Remarks to the Author:

This methodological study validates the use of micro-CT and MRI for in vivo localization of chronic implants in mice. This technique is not a substitute for confirming the targeted location using well-established ex vivo methods, but it has the potential to save countless hours of effort spent training or recording from subjects that will not yield useful data. The authors demonstrate that aligning post-surgical CT scans with pre-surgical CT and MRI can be used to identify the location of fiber optic cables or electrodes to within 100 microns of the histologically validated site. This level of accuracy is sufficient for targeting most deep structures, given the typical spread of light and electrode bundles in tissue. Crucially, the authors are attuned to the potential dangers of micro-CT-related radiation exposure. They perform experiments to establish the device settings that yield sufficiently high

resolution and contrast without any noticeable health effects. They also confirm that X-ray exposure does not cause a marked decrement in behavioral task performance, although this conclusion is based on a small sample size. Overall, this is an important contribution to the field of in vivo mouse physiology, and I only recommend minor revisions:

The results in figure 6 average over 9 different target locations distributed across the HDB, medial septum, and VTA. Because there are so few measurements, it would be helpful to plot the individual data points, colored by target region, in addition to the mean \pm std.

The authors describe a previous study in which 13 mice were excluded following histological reconstructions that indicated the electrode bundle was not targeted accurately. The present method could have prevented these mice from completing the entire experiment, but were there any additional mice that could have been salvaged by knowing the actual recording location? That is, could this method have increased the N for this study, in addition to removing mice that were not useful?

The method described here relies on rodent-compatible micro-CT and MRI machines, relatively expensive devices that are not likely to be found in the average neuroscience department. Are there any insights that can be drawn from this study that would help those without access to these tools? For example, is the systematic inaccuracy in the A/P targeting likely to be consistent across labs using similar stereotaxic hardware?

A detailed analysis of the correspondence between structural MRI and micro-CT collected from the same individuals is beyond the scope of this manuscript. However, because these co-registered datasets are so rare, I strongly recommend making the volumes readily available to the community, via Figshare or a similar data repository. It is likely that valuable insights can be gleaned from this dataset, and it should be made easily accessible to other researchers.

Change "silicone" to "silicon" when referring to silicon probes (lines 179, 182, 183, 287, 643)

Line 348: Change "decent" to "descent"

Reviewer #3:

Remarks to the Author:

Király et al., proposes a potential method for In Vivo Localization of Deep Brain Implants in Mice. The authors mainly propose a method combining the information about bone landmarks provided by micro-CT scanning with the soft tissue contrast of the MRI to precisely localize electrodes and optic fibers in mice in vivo.

A very similar approach has already been proposed for rats (Borg et al., 2015; Rangarajan et al., 2016), the novelty comprises the application of this method for the mouse.

They use high-resolution (19 μ m) micro-CT imaging to localize the brain implants with respect to bone landmarks. As CT does not provide soft tissue contrast, they performed structural MRI scans (1T permanent magnetic field) and merged them with the micro-CT images.

The paper could be of interest to others in the field, but some points should be clarified and the presentation of the results should be reorganized in a revised manuscript.

Major comments:

- 1.) Figure 1. – A more detailed in vivo localization workflow should be depicted.
- 2.) In the beginning of results section, they describe the pre-operative micro-CT measurements at 35 μm resolution then they describe the post-operative CT scanning at 19 μm resolution. Only at the very end of the results section they describe the “Optimization of image quality and radiation dose”. This should be described earlier to explain the difference in resolution in the pre- and post-operative CT scanning.
- 3.) In the methods and results section more details should be added how they obeyed the ALARA principle (‘as low as reasonably achievable’) for radiation protection. More specifics should be added about the particular animal experiments, provide the name of the ethical review committee that has approved the current experiments in this study and any relevant license or protocol numbers and if they followed the ARRIVE Guidelines.
- 4.) Figure 2. – The Paxinos atlas coordinates should be added to the sagittal, coronal and horizontal planes. The bone structures used for leveling and co-registration should be marked.
- 5.) Figure 4. – The localization of multiple-targeting implants, silicon probes and optic fibers, contains only CT images. Here also, the Paxinos atlas coordinates should be added. This figure and the corresponding results section in the current state should precede Figure 3 and the CT-MRI fusion results section.
- 6.) The claim that they could localize the silicone probe tip is not convincing. It is not plausible even with using gold-standard histology.
- 7.) They do not attempt to localize the silicone probe tip with the gold-standard histology, only tetrodes.
- 8.) Figure 3. – The Paxinos atlas coordinates should be added.
- 9.) The exact dimensions of the 8 tetrode bundle should be provided not just the 12.7 μm single wire diameter.
- 10.) Regarding the “systematical error in the stereotaxic leveling”, the Methods state: 410-412 „The mouse was placed in a stereotaxic frame (David Kopf Instruments, Tujunga, CA, USA) and the skull was levelled. The skin was incised, the skull was cleaned and a cranial window was opened above the target area...”. The skull was levelled before the skin was incised?!
- 11.) About the “Optimization of image quality and radiation dose”, the results state: 290-292 „the highest radiation dose, we observed a slight loss of hair and skin irritation around the neck of the animals after 2-3 weeks”. The eyes of the mice were examined (Barnard et al., 2018)?
- 12.) How the animals were anaesthetized during scanning? If the same as during Stereotaxic Surgery (25 mg/kg xylazine and 125 mg/kg 406 ketamine) ketamine with Low-Dose Ionizing Radiation could adversely influence CA1 neuronal structure in mice hippocampi (Hladik et al., 2019).

References

Barnard, S. G. R., Moquet, J., Lloyd, S., Ellender, M., Ainsbury, E. A., & Quinlan, R. A. (2018). Dotting the eyes: mouse strain dependency of the lens epithelium to low dose radiation-induced DNA damage. *Int J Radiat Biol*, 94(12), 1116-1124. doi:10.1080/09553002.2018.1532609

Borg, J. S., Vu, M. A., Badea, C., Badea, A., Johnson, G. A., & Dzirasa, K. (2015). Localization of Metal Electrodes in the Intact Rat Brain Using Registration of 3D Microcomputed Tomography Images to a Magnetic Resonance Histology Atlas. *eNeuro*, 2(4). doi:10.1523/ENEURO.0017-15.2015

Hladik, D., Buratovic, S., Von Toerne, C., Azimzadeh, O., Subedi, P., Philipp, J., . . . Tapio, S. (2019). Combined Treatment with Low-Dose Ionizing Radiation and Ketamine Induces Adverse Changes in CA1 Neuronal Structure in Male Murine Hippocampi. *Int J Mol Sci*, 20(23). doi:10.3390/ijms20236103

Rangarajan, J. R., Vande Velde, G., van Gent, F., De Vloo, P., Dresselaers, T., Depypere, M., . . . Maes, F. (2016). Image-based in vivo assessment of targeting accuracy of stereotactic brain surgery in experimental rodent models. *Sci Rep*, 6, 38058. doi:10.1038/srep38058

Rebuttal letter: In Vivo Localization of Chronic Brain Implants in Mice

Structure:

Reviewer comments: black, italic

Our replies: blue

We would like to thank the Reviewers for their constructive comments and positive feedback, finding ‘the study to be meticulously conducted and clearly reasoned and written’ (Reviewer #1), ‘of interest to others in the field’ (Reviewer #3) and ‘an important contribution to the field of in vivo mouse physiology’ (Reviewer #2).

Suggestions from all three Reviewers greatly helped us improve our study.

1. To broaden the range of applications, we improved *in vivo* localization, combining micro-CT with translational-field (3T) MRI imaging, and performed new experiments where hippocampal implants are localized with an accuracy sufficient to provide layer information.
2. To respond to the concern of potentially limited availability of imaging equipment, we also explored localization of metal-free implants based on 3T MRI images only. Thus, we now include localization methods based on micro-CT only, MRI only and CT-MRI fusion, increasing the flexibility of applications.
3. We improved text and figures based on the Reviewers’ requests and decided to make the co-registered CT and MRI images freely available for further use.

Our point-by-point responses are laid out in details below.

Response to Reviewer #1

Kiraly et al presented a new platform to localize deep brain implants in mice. This platform is based on micro-CT and MRI, which respectively provides bone structural information and soft tissue contrast information. By registering pre-surgical micro-CT and MRI, and align the images to post-surgical micro-CT including the implants, the authors report location precision comparable to histological reconstruction. The obvious benefit is that CT/MRI based location is non-invasive and can be conducted repeatedly. Moreover, the localization information can be used to guide the continuation or early termination of experiments, and substantially increase the efficiency of experiments involving deep brain implants such as electrodes or optic fibers.

Using this new localization platform, the authors also identified systematic targeting errors, especially in the AP (antero-posterior) direction. This is likely due to insufficient leveling between bregma and lambda, leading to a slight tilt of the skull, and hence introduces an unwanted angle of the implant in the AP direction. This problem is amplified in mice because the size of the skull is smaller, and thus the same the leveling error would lead to a bigger angular error (compared to the same error made in a larger animal such as the rat).

The authors also characterized the spatial resolution of micro-CT under different imaging parameters and the corresponding radiation exposure dose, and demonstrated that such exposure doses were safe and did not adversely affect animals' behavior.

Overall, I find the study to be meticulously conducted and clearly reasoned and written. The results are sound. The idea is not original (for example Borg et al 2015 cited in this manuscript), but the current implementation is substantially better and provides conceptual insights about reasons for targeting error, as well as recommended parameters for using this method.

We thank the Reviewer for the accurate summary and positive assessment of our manuscript.

A minor concern though is that the results are relevant to only a small group of researchers who target deep brain structures where the localization errors are more significant. And perhaps the most relevant audience are those who implant moveable tetrode (which are much smaller than optic fiber implants) that need to travel a long distance before reaching the target.

We thank the Reviewer for raising this important point. We indeed focused on relatively small, deep structures, because those are often notoriously hard to hit by stereotaxic surgery in mice. However, based on the Reviewer's note, we conducted additional experiments with hippocampal implants (n = 5; 2 silicon probe and 3 fiber optic implants). While cortical and hippocampal targets are typically much easier to hit, most studies require precise layer-specific targeting.

In n = 3 mice, we assessed the utility of the CT-MRI-fusion-based *in vivo* localization for hippocampal silicon probe and optic fiber implants and showed that *in vivo* localization may aid layer-specific targeting. In another n = 2 mice implanted bilaterally with optic fibers of different diameter, we tested whether translational-field (3T) MRI imaging alone can localize metal-free optic fiber implants and found that it was possible to achieve a precision sufficient for most applications. These experiments are presented in a new section of Results ('Localization of hippocampal implants using high SNR MRI imaging', Fig.5-6). Therefore, we argue that this technique will also be of interest to those researchers who target specific layers of the neocortex or the hippocampus.

We agree with the Reviewer that *in vivo* localization may be critical for tetrode implants, while optic fiber implants for optogenetic manipulation of neuron types with restricted expression patterns may be easier, since optogenetic manipulations are rarely limited by light intensity. At the same time, we encountered several possible problems of fiber optic and silicon probe implants that may make *in vivo* localization useful for these techniques. (i) Optogenetic manipulations are often targeted at neuron types that are not restricted to the target zone (e.g. GABAergic, glutamatergic, parvalbumin-expressing), in which case precise positioning of the optic fibers is important; (ii) large, stiff optic fibers may introduce lesions in the target zone if implanted too deep, compromising the interpretation of the experiment; (iii) fiber positioning in fiber photometry experiments is important in order to gain a good quality signal from the targeted population; (iv) silicon probes often bend on white matter tracts, complicating precise aiming; (v) as Reviewer #3 pointed out, localization of silicon probe tips is difficult with histology; since CT images visualize the probe from the base, the known length of the probe provides full verification of tip location (see also response to point 6 of Reviewer #3). We now include these arguments in the Discussion section (lines 441-450).

Also, the detailed design of the specialized isoflurane mask that maintains head fixation during micro-CT/MRI imaging should be made available to other scientists to aid the dissemination and replication of this method.

We provided reference to open source animal holders with isoflurane mask, appropriate for our *in vivo* localization method (line 84). We used a similar isoflurane mask that was sold with the MRI and CT machines, but the design files of this holder are protected under IP rights and we were not permitted to share them.

Response to Reviewer #2

This methodological study validates the use of micro-CT and MRI for in vivo localization of chronic implants in mice. This technique is not a substitute for confirming the targeted location using well-established ex vivo methods, but it has the potential to save countless hours of effort spent training or recording from subjects that will not yield useful data. The authors demonstrate that aligning post-surgical CT scans with pre-surgical CT and MRI can be used to identify the location of fiber optic cables or electrodes to within 100 microns of the histologically validated site. This level of accuracy is sufficient for targeting most deep structures, given the typical spread of light and electrode bundles in tissue. Crucially, the authors are attuned to the potential dangers of micro-CT-related radiation exposure. They perform experiments to establish the device settings that yield sufficiently high resolution and contrast without any noticeable health effects. They also confirm that X-ray exposure does not cause a marked decrement in behavioral task performance, although this conclusion is based on a small sample size.

We thank the Reviewer for the positive assessment of our study. We increased the sample size of the behavioral experiment from $n = 2$ to $n = 7$ mice (Fig. 10c). This increased sample provided the same result.

Overall, this is an important contribution to the field of in vivo mouse physiology, and I only recommend minor revisions:

The results in figure 6 average over 9 different target locations distributed across the HDB, medial septum, and VTA. Because there are so few measurements, it would be helpful to plot the individual data points, colored by target region, in addition to the mean \pm std.

We increased the sample size to $n = 12$ and included the individual data points, as suggested by the Reviewer. The individual data of all experiments are also presented in Supplementary Figure 6 and 7.

The authors describe a previous study in which 13 mice were excluded following histological reconstructions that indicated the electrode bundle was not targeted accurately. The present method could have prevented these mice from completing the entire experiment, but were there any additional mice that could have been salvaged by knowing the actual recording location? That is, could this method have increased the N for this study, in addition to removing mice that were not useful?

Although we cannot unequivocally determine whether some of the mice could have been salvaged in the study we analyzed post hoc in the manuscript, it happens regularly in our experience that the implant is dorsal to the target, and adjusting the depth of the implanted electrodes with micro-drives greatly aids the experiment. Even if it is conceivable that the target could have been reached by blind slow stepwise descent through days, this not only takes additional time but recording quality (and chance of survival) decreases gradually in the process. Current Fig. 7 depicts a case where the implant was dorsal to the target and CT could provide the necessary information for proper depth adjustment. We added a note on this topic in the Discussion (lines 399-402).

The method described here relies on rodent-compatible micro-CT and MRI machines, relatively expensive devices that are not likely to be found in the average neuroscience department. Are there any insights that can be drawn from this study that would help those without access to these tools? For example, is the systematic inaccuracy in the A/P targeting likely to be consistent across labs using similar stereotaxic hardware?

We thank the Reviewer for raising this point. We agree that identification of systematic implantation errors may be rather general and not specific to certain operators. We found that multiple experimenters err on AP leveling, probably due to the uncertainty of Bregma and Lambda points – particularly the latter, because there is a level difference between the occipital and parietal plates in mice at Lambda, and there is no clear guidance about which point serves as best reference. As Reviewer #1 pointed out, the introduced error may be amplified in mice compared rats due to the smaller size of the skull, leading to a larger error in the implanting angle. We added a discussion on this topic (lines 422-430).

While we realized that micro-CT and MRI machines are often not in routine use in neuroscience labs, our internet searches of university web sites indicated that many university centers have imaging core facilities that possess these pieces of equipment, regularly used by multiple medical research fields. (Indeed, 3 out of the 4 medical universities in Hungary offer these imaging modalities.) Nevertheless, to make *in vivo* localization more accessible, we evaluated localization accuracy both based on micro-CT alone (equipment usually affordable at the level of individual labs) and based on CT-MRI fusion. To further increase flexible use, we now explored the possibility of localization based on MRI only (newly added Result section ‘Localization of hippocampal implants using high SNR MRI imaging’, Fig.5-6, Supplementary Figure 4; see also Discussion, lines 458-466).

A detailed analysis of the correspondence between structural MRI and micro-CT collected from the same individuals is beyond the scope of this manuscript. However, because these co-registered datasets are so rare, I strongly recommend making the volumes readily available to the community, via Figshare or a similar data repository. It is likely that valuable insights can be gleaned from this dataset, and it should be made easily accessible to other researchers.

We agree with the Reviewer. The data set will be made available via Figshare at the time of publication.

Change “silicone” to “silicon” when referring to silicon probes (lines 179, 182, 183, 287, 643)

Line 348: Change “decent” to “descent”

We fixed these typos.

Response to Reviewer #3

Király et al., proposes a potential method for In Vivo Localization of Deep Brain Implants in Mice. The authors mainly propose a method combining the information about bone landmarks provided by micro-CT scanning with the soft tissue contrast of the MRI to precisely localize electrodes and optic fibers in mice in vivo.

A very similar approach have already proposed for rats (Borg et al., 2015; Rangarajan et al., 2016), the novelty comprise the application of this method for the mouse.

They use high-resolution (19 μm) micro-CT imaging to localize the brain implants with respect to bone landmarks. As CT does not provide soft tissue contrast, they performed structural MRI scans (1T permanent magnetic field) and merged them with the micro-CT images.

The paper could be of interest to others in the field, but some points should be clarified and the presentation of the results should be reorganized in a revised manuscript.

We thank the Reviewer for the constructive comments, based on which we believe the manuscript could be significantly improved.

Major comments:

1.) *Figure 1. – A more detailed in vivo localization workflow should be depicted.*

We improved the workflow figure.

2.) *In the beginning of results section, they describe the pre-operative micro-CT measurements at 35 μm resolution then they describe the post-operative CT scanning at 19 μm resolution. Only at the very end of the results section they describe the “Optimization of image quality and radiation dose”. This should be described earlier to explain the difference in resolution in the pre- and post-operative CT scanning.*

Thank you for noting this. We added the explanation (lines 85-87).

3.) *In the methods and results section more details should be added how they obeyed the ALARA principle (‘as low as reasonably achievable’) for radiation protection. More specifics should be added about the particular animal experiments, provide the name of the ethical review committee that has approved the current experiments in this study and any relevant license or protocol numbers and if they followed the ARRIVE Guidelines.*

We provided the relevant information and attached an ARRIVE checklist.

We expanded the corresponding section on ‘Animals’ in Methods, which now reads

‘All experiments were approved by the Institutional Animal Care and Use Committee and the Committee for the Scientific Ethics of Animal Research of the National Food Chain Safety Office (PE/EA/675-4/2016,

PE/EA/1212-5/2017, PE/EA/864-7/2019) and were performed according to the guidelines of the institutional ethical code and the Hungarian Act of Animal Care and Experimentation (1998; XXVIII, section 243/1998, renewed in 40/2013) in accordance with the European Directive 86/609/CEE and modified according to the Directives 2010/63/EU. BK, IH, KSz and BH each hold a certificate course diploma of Advanced Radiation Protection (with certification numbers H04/2017, B-2019/54, OSSKI-2014-ÁK-369-21, SUVE-B-059/2008), granted by the Budapest University of Technology and Economics or the Semmelweis University, Budapest according to the Hungarian Act CXVI of 1996 on Atomic Energy ('Atomic Act'), granted based on the permission of the National Public Health Service of Hungary.'

We added a paragraph to this section on the ALARA principle (lines 483-496) and expanded the Pre-operative scanning section to provide further information on how the ALARA principle was implemented (lines 556-560).

4.) Figure 2. – The Paxinos atlas coordinates should be added to the sagittal, coronal and horizontal planes. The bone structures used for leveling and co-registration should be marked.

We added the Paxinos coordinates to the images (numbers in yellow) and provided an enlarged view of CT images with the bone structures that were used for leveling marked in Supplementary Figure 1. The same structures are also marked in Fig. 2b-c by yellow dashed lines connecting corresponding structures.

5.) Figure 4. – The localization of multiple-targeting implants, silicon probes and optic fibers, contains only CT images. Here also, the Paxinos atlas coordinates should be added. This figure and the corresponding results section in the current state should precede Figure 3 and the CT-MRI fusion results section.

We reorganized the Results as suggested by the Reviewer. Additionally, we now provide pre-operative MRI images for silicon probes and optic fibers in Fig. 5a and Supplementary Fig. 3 and post-operative MRI images of fiber optics implants in Fig. 6. The Paxinos coordinates has been added to the image planes.

6.) The claim that they could localize the silicone probe tip is not convincing. It is not plausible even with using gold-standard histology.

Thank you for raising this point. We confirmed silicon probe tip positions using the following procedure. Since the entire probe is visible in the CT image directly from its base, the known length of the probe provides a handle to verify the image-based tip localization. We compared the image-based tip coordinates with the expected coordinates based on the published dimensions of the probe and found CT-based tip localization very accurate. We demonstrate this in Supplementary Figure 2 and included it in the Results section (lines 135-141). This made us realize that *in vivo* localization of silicon probes may have the added benefit that, in contrast to histological reconstruction, it is possible to determine the tip position using micro-CT imaging.

7.) They do not attempt to localize the silicone probe tip with the gold-standard histology, only tetrodes.

We added histological assessment of silicon probe tip coordinates in Supplementary Figure 5 and included 2 animals implanted with silicon probes in the analysis of the accuracy of *in vivo* localization (Fig. 8 and Supplementary Figure 7).

8.) *Figure 3. – The Paxinos atlas coordinates should be added.*

Paxinos coordinates were added to the images.

9.) *The exact dimensions of the 8 tetrode bundle should be provided not just the 12.7 µm single wire diameter.*

We provided Table 1. with the dimensions of implants used for our study (including tetrode bundle) and expanded the Stereotaxic Surgery section of Methods with the following sentence: 'Tetrode bundles covered an approximately cylindrical volume in the tissue with non-uniform gaps between the tetrodes, with an average diameter of 0.34 ± 0.07 mm based on the CT images.' (lines 516-518)

10.) *Regarding the “systematical error in the stereotaxic leveling”, the Methods state: 410-412 „The mouse was placed in a stereotaxic frame (David Kopf Instruments, Tujunga, CA, USA) and the skull was levelled. The skin was incised, the skull was cleaned and a cranial window was opened above the target area...”. The skull was levelled before the skin was incised?!*

Thank you. The error was corrected.

11.) *About the “Optimization of image quality and radiation dose”, the results state: 290-292 „the highest radiation dose, we observed a slight loss of hair and skin irritation around the neck of the animals after 2-3 weeks”. The eyes of the mice were examined (Barnard et al., 2018)?*

Thank you for raising this point. The eyes were indeed examined, and we observed no signs of cataract or any other adverse effects in the eye lenses. This information is now included in the manuscript (lines 336, 454).

12.) *How the animals were anaesthetized during scanning? If the same as during Stereotaxic Surgery (25 mg/kg xylazine and 125 mg/kg 406 ketamine) ketamine with Low-Dose Ionizing Radiation could adversely influence CA1 neuronal structure in mice hippocampi (Hladik et al., 2019).*

Mice were anesthetized with isoflurane during scanning, as adopted from other medical fields (see Methods, line 551).

Reviewers' Comments:

Reviewer #1:

Remarks to the Author:

The authors have adequately addressed all my concerns. The revised manuscript is much improved. I only noticed one minor issue that requires editing: In Figure 3B, the AP/ML/DV coordinates were not added to the panels.

Reviewer #2:

Remarks to the Author:

The authors have successfully addressed all of my concerns. The revised manuscript is more comprehensive, with many beautiful figures. I am pleased to hear the authors plan on releasing their volumetric datasets, which will be a valuable resource for the community.

Reviewer #3:

Remarks to the Author:

My concerns have been reasonably addressed by either the new experiments or improved text and figures.

I recommend publication.

REVIEWERS' COMMENTS:

Reviewer #1 (Remarks to the Author):

The authors have adequately addressed all my concerns. The revised manuscript is much improved. I only noticed one minor issue that requires editing: In Figure 3B, the AP/ML/DV coordinates were not added to the panels.

We thank the Reviewer for his/her constructive comments and for finding our revised manuscript much improved. The coordinates were not added to Figure 3b, because the image was rotated from the canonical axes for visualization purposes, as these silicon probes are too thin to be properly visible without rotation at normal image resolutions. We updated the figure legend to clarify this.

Reviewer #2 (Remarks to the Author):

The authors have successfully addressed all of my concerns. The revised manuscript is more comprehensive, with many beautiful figures. I am pleased to hear the authors plan on releasing their volumetric datasets, which will be a valuable resource for the community.

We thank the Reviewer for his/her constructive comments and for the positive assessment of our study.

Reviewer #3 (Remarks to the Author):

My concerns have been reasonably addressed by either the new experiments or improved text and figures.

I recommend publication.

We thank the Reviewer for his/her constructive comments and for recommending publication of our manuscript.